# Envelope reconstruction of speech and music highlights stronger tracking of speech at low frequencies

**Nathaniel J. Zuk**[1,2,3,4,5¤a]*, **Jeremy W. Murphy**[1¤b], **Richard B. Reilly**[2,3,6], **Edmund C. Lalor**[1,4,5]

**1** Department of Electronic & Electrical Engineering, Trinity College, The University of Dublin, Dublin, Ireland, **2** Department of Mechanical, Manufacturing & Biomedical Engineering, Trinity College, The University of Dublin, Dublin, Ireland, **3** Trinity College Institute of Neuroscience, Trinity College, The University of Dublin, Dublin, Ireland, **4** Department of Biomedical Engineering, University of Rochester, Rochester, New York, United States of America, **5** Del Monte Institute of Neuroscience, University of Rochester Medical Center, Rochester, New York, United States of America, **6** Trinity Centre for Biomedical Engineering, Trinity College, The University of Dublin, Dublin, Ireland

¤a Current address: Edmond & Lily Safra Center for Brain Sciences, Hebrew University, Jerusalem, Israel
¤b Current address: Department of Neuroscience, Brown University, Providence, Rhode Island, United States of America
* nathaniel.zuk@mail.huji.ac.il

**Data Availability Statement:** The EEG data cannot be shared publicly for ethical reasons. The original ethical approval obtained at the time the data was collected (2014-2015) did not allow us to make the data publicly available. To access the data,

## Abstract

The human brain tracks amplitude fluctuations of both speech and music, which reflects acoustic processing in addition to the encoding of higher-order features and one's cognitive state. Comparing neural tracking of speech and music envelopes can elucidate stimulus-general mechanisms, but direct comparisons are confounded by differences in their envelope spectra. Here, we use a novel method of frequency-constrained reconstruction of stimulus envelopes using EEG recorded during passive listening. We expected to see music reconstruction match speech in a narrow range of frequencies, but instead we found that speech was reconstructed better than music for all frequencies we examined. Additionally, models trained on all stimulus types performed as well or better than the stimulus-specific models at higher modulation frequencies, suggesting a common neural mechanism for tracking speech and music. However, speech envelope tracking at low frequencies, below 1 Hz, was associated with increased weighting over parietal channels, which was not present for the other stimuli. Our results highlight the importance of low-frequency speech tracking and suggest an origin from speech-specific processing in the brain.

## Author summary

The time-varying amplitude of sounds such as speech and music provides information about phrasing and rhythm, and previous research has shown that the brain continuously tracks these variations. Is a common neural mechanism responsible for tracking both sounds? We used a technique that reconstructs these amplitude fluctuations from the fluctuations in recorded EEG to quantify the strength of neural tracking. Our hypothesis was

interested researchers can contact the Ethics Committee for the School of Psychology at Trinity College Dublin (psych.ethics@tcd.ie)

**Funding:** NZ, RR, and EL were supported by Science Foundation Ireland Career Development Award (CDA/15/3316). (https://www.sfi.ie/) JM and EL were also supported by an Irish Research Council Starter Research Project Grant (RPG2013-1). (https://research.ie/) Additional support was provided to NZ and EL by the Del Monte Institute for Neuroscience at the University of Rochester. (https://www.urmc.rochester.edu/del-monte-neuroscience.aspx) The funders had no role in study design, data collection and analysis, decision to publish, or preparation of the manuscript.

**Competing interests:** The authors have declared that no competing interests exist.

that neural tracking for music would match speech in a narrow frequency range associated with syllables and musical beats. Though our results did suggest a common mechanism involved in tracking both speech and music at these higher frequencies, we instead found that speech was tracked better than the rock, orchestral, and vocals stimuli at all of the frequencies we examined. Moreover, low-frequency fluctuations at the rate of phrases were associated with increased EEG weightings over parietal scalp, which did not appear during the neural tracking of music. These results suggest that, unlike syllable- and beat-tracking, phrase-level tracking of amplitude variations in speech may be speech-specific.

## Introduction

Sound carries information in speech and music across a wide range of time scales, and theoretically the brain must have some parsing mechanism that operates on the sound prior to extracting information. There is considerable debate about how speech and music are parsed by the brain, but it is fairly clear that the sound's rhythmic structure enables speech decoding [1], facilitates pattern recognition [2], and affects discrimination of orthogonal acoustic attributes [3,4]. Fluctuations in the envelopes of speech and music capture rhythmic structure, so if the brain is responding to this structure, brain activity should also be tracking their envelopes.

Indeed, past research has shown that brain signals track envelopes for both speech (for reviews see: [5,6]) and music [7–9]. Furthermore, the strength of envelope tracking can be affected by the listener's locus of attention [10–13], level of comprehension [14,15], and perceived musical meter [16–18]. Thus, rather than simply representing a response to the acoustical structure, envelope tracking may include neural mechanisms that respond to more complex, non-acoustic features which correlate with the envelope but do not represent the stimulus envelope per se [8,19–22].

The mechanisms that produce this envelope tracking are still hotly debated, however, and the various interpretations typically stem from the method of analysis used in each study. One theory posits that neural oscillations are tuned to particular frequencies relevant for parsing the speech or music into these discrete units [23,24]. For speech, the phase consistency of the neural responses observed at theta band (4–8 Hz) and delta band (< 4 Hz) correspond roughly to syllabic and phrase rates respectively. In another theory, the neural tracking of the envelope measured with EEG and MEG is represented by its convolution with an evoked response [25,26]. The evoked response has been shown to respond particularly to envelope edges, which mark vowel-nucleus onsets and are highly correlated with syllable onsets in speech [20,27]. Yet it is not clear if the two mechanisms are irreconcilably different. Phase consistency can come about if characteristic evoked responses occur at transient, regular times in the sound [27]. Likewise, an evoked response model that captures envelope tracking acts like a filter [28] and, via regularization, focuses on lower frequencies that overlap with the frequencies exhibiting the largest phase consistency. Evoked-response-type models have been used to quantify and compare the strength of neural tracking in theta and delta bands [14,19,29], which also suggests that, in spite of the differing theories for the mechanisms of neural tracking, the distinct results across studies could be related. This can be examined further by computing models responsible for tracking various frequency bands of the envelope and looking at the corresponding evoked responses captured by those models.

Neural tracking in the theta band appears to be strongest for speech and music compared to other natural sounds [30], yet it remains to be determined if the underlying mechanisms of envelope tracking are distinct between speech and music. A model-based approach can test

this directly by examining how stimulus-response models differ across stimulus types. Unfortunately, comparing model-based quantifications of speech and music neural tracking is complicated by the different spectral profiles between the two types of sounds. Additionally, neural tracking is also affected by feedback mechanisms relating to non-acoustic features [21,22,31] or one's cognitive state [10,11,16,32,33]. But to compare these "higher-level" effects between speech and music in naturalistic stimuli it is useful to first isolate differences in neural tracking that cannot be explained by envelope spectral differences alone.

In this study, we apply a variant on linear modeling to focus analysis on a narrow range of frequency bands. We then compare reconstruction accuracies for various shifts of the frequency range in order to understand precisely how EEG tracks the signal envelope of speech and music during passive listening. We hypothesized that speech and music would have different frequency bands at which their envelopes are tracked best, and by using linear modeling, we could observe the spatiotemporal patterns of the responses that are most relevant for envelope tracking in each respective frequency region.

## Results

In our experiment, 16 subjects (7 female; ages 18–44, median 22) passively listened to four types of stimuli, interleaved across trials: rock songs in full, excerpts from orchestral songs, the vocals isolated from the rock songs, and several segments of an audiobook (called "speech" stimuli in the figures). Each subject listened to 6–7 of the 10 possible stimuli available for each stimulus type (see S1 Table for details on the stimuli). We will first step through the model construction and optimization procedure. Then, we will show the differences in neural tracking we observed and examine the spatiotemporal EEG responses that gave rise to these differences in neural tracking.

### Envelope reconstruction with PCA and basis spline transformation

Our initial goal was to quantify how well EEG tracked the music envelopes in comparison to speech. Previous research had shown that EEG tracks envelope fluctuations in speech and music particularly above 2 Hz [7,34,35]. Based on this prior work, we expected that music envelope tracking would be comparable to speech when reconstructions are focused on a narrower bandwidth of frequencies above 2 Hz. Additionally, speech and vocals might show low frequency tracking of envelopes, below 1 Hz, where the energy in orchestral and rock envelopes is comparatively low (Fig 1A and 1B; see also S1 Fig). On the other hand, if we assume that the neural signal in the EEG is directly proportional to the dB envelope, we might expect speech tracking to be largest across all modulation frequencies, followed by vocals, then orchestral, and then rock music (Fig 1B). With this in mind, we decided to use an approach to envelope reconstruction that focused on narrow bands of modulation frequencies. This would allow us to identify if differences in envelope reconstruction that we observe using the broadband envelope (see S2 Fig) are due to differences in the modulation frequencies that are most strongly tracked. Alternatively, envelope tracking of speech could be better because of the relative variance in the modulations of the dB envelope.

We used a method of modeling that allowed us to focus on envelope tracking in narrow frequency bands. We chose to implement this filtering stage within the model itself, rather than as a separate stage of preprocessing, so that we could use an identical model architecture and common hyperparameters to investigate each frequency range (Fig 2A). First, the moving average of both the stimulus envelope and EEG, using a window size equal to the maximum EEG delay in the model, was removed in order to remove effects of neural tracking outside of the model's spectral range (see S2C Fig). Next, the EEG data were spatially transformed into

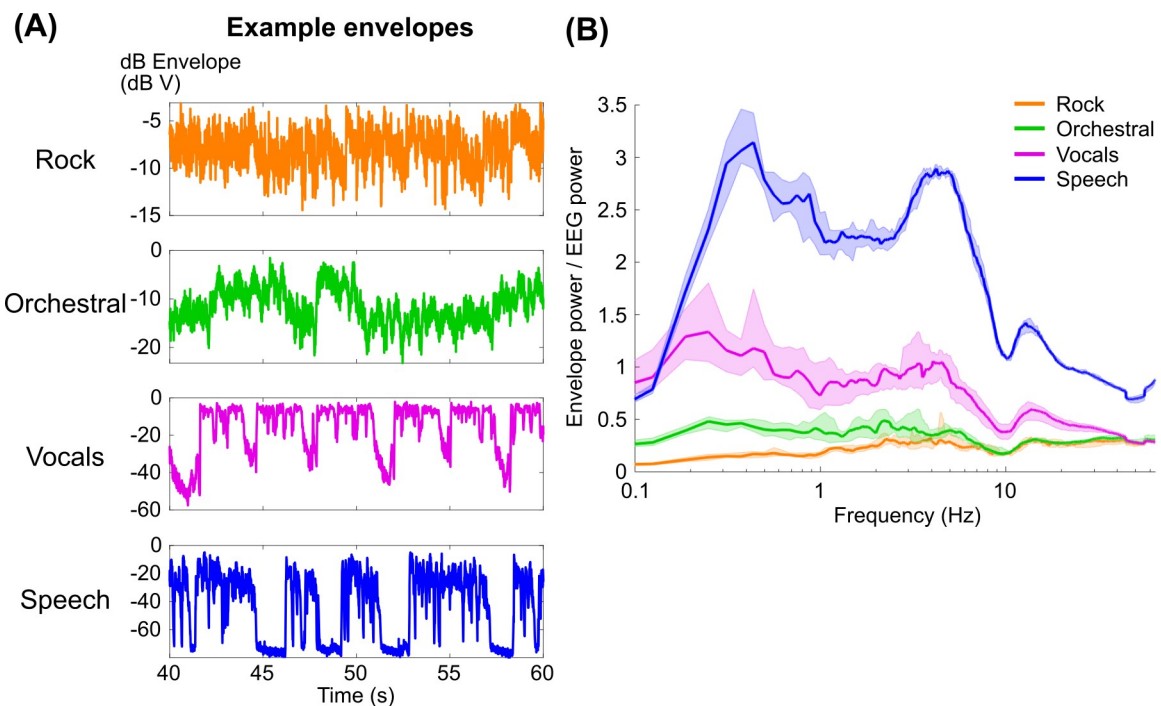

**Fig 1.** (A) 20 seconds of example dB envelopes for each stimulus type. (B) Power spectra of the dB envelopes using Welch's method with a 16 s Hamming window and half-overlap after normalizing by the average EEG spectrum for all subjects (see Methods). Lines indicate the median across stimuli of each type, and shaded regions indicate 95% quantiles of the distribution of 1000 bootstrapped median values.

orthogonal components using principal components analysis (PCA) in order to remove spatial collinearity in the EEG. Lastly, before fitting the model, the time-delayed EEG signal was transformed into a basis of cubic splines. Cubic splines are smooth functions constructed based on a sequence of knots. By collating these splines across the range of delays, the splines can be used to interpolate any delayed EEG with a smoother, lower-frequency representation. The number of basis splines can be equivalently defined by the sampling frequency of the evenly distributed knots, which also relates to the lowpass cutoff frequency of the splines (see legend in Fig 2B). Overall, for a specific window size, this model contains two hyperparameters: 1) the number of principal components to retain, and 2) the number of basis splines within the model window size.

To optimize these hyperparameters, we fitted the model using the Natural Speech dataset which is freely available online [36]. In this dataset, subjects (N = 19) listened to 20 trials of an audiobook that were approximately three minutes long. In order to identify the optimal hyperparameters, we compared the PCA & spline model performance to a regularized model using ridge regression, which is commonly used for speech envelope reconstruction [13,37–39]. Specifically, we compared the performance of the two models on each trial by training and testing the models using a leave-one-trial-out procedure. When a small number of principal components and splines are included, the model tends to underperform compared to models with more parameters (Fig 2B). Because no regularization was applied to the PCA & spline model, it also underperforms when all principal components are included. The model containing 64 principal components and 19 splines (or equivalently a spline knot sampling rate of 32 Hz) exhibited the best performance and was the only hyperparameter pair whose performance was not significantly different than the model with regularization (Fig 2B) (Wilcoxon rank-sum test, $p > 0.05$). The optimal sampling of spline knots constrained the frequency content of the time-delayed

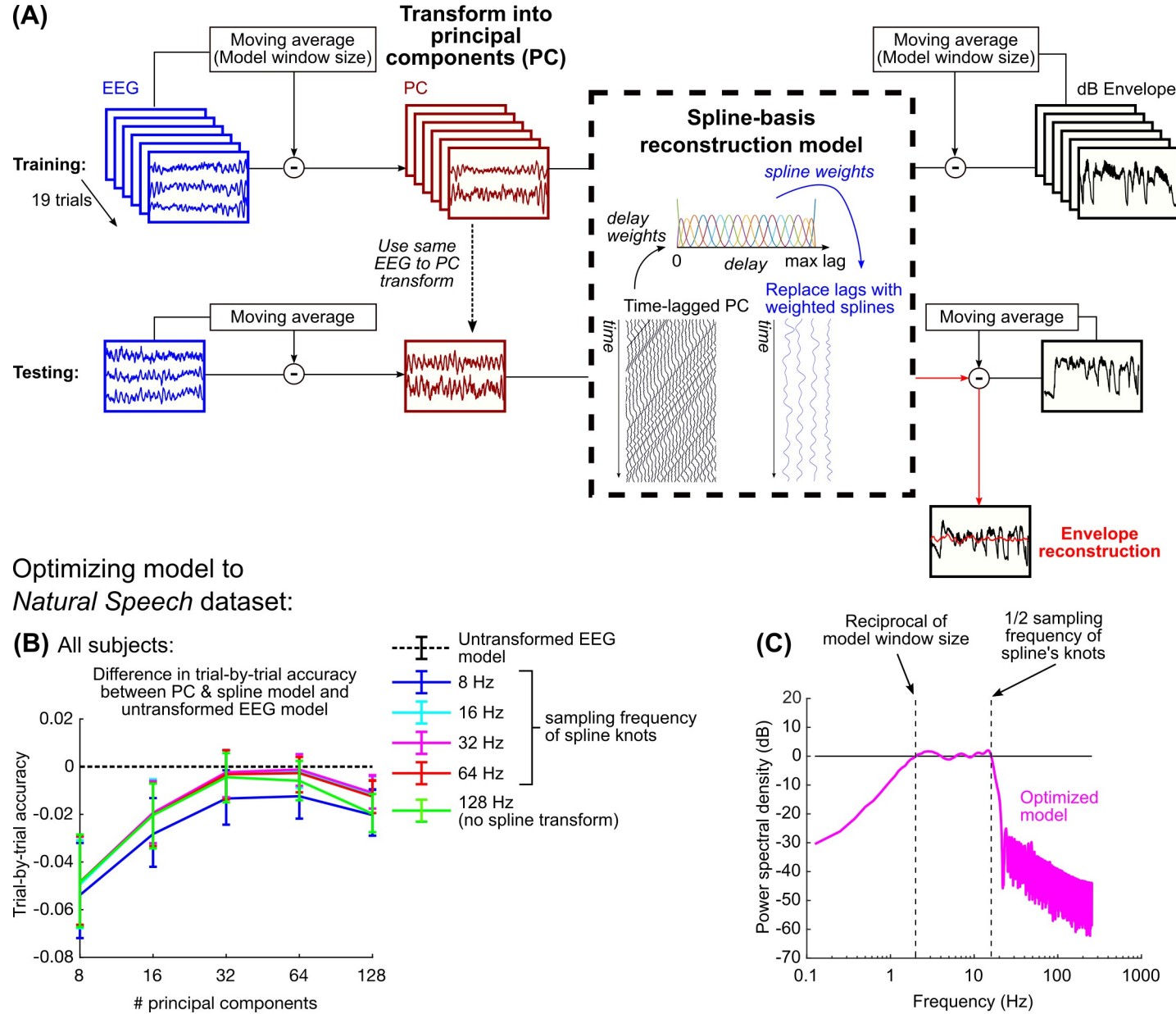

**Fig 2.** (A) Diagram of the stages for fitting the PCA & spline model. Throughout this study, the model is trained on all trials with one left out and tested on the left-out trial. (B) The difference in reconstruction accuracy (Pearson's r) between the PCA & spline method and a standard approach based on regularized linear regression for each of the left-out trials was then examined; negative values mean that the standard approach performs better. Error bars show the interquartile range of the reconstruction accuracy differences. Of all hyperparameter pairs, only 64 PCs and spline knots sampled at 32 Hz exhibited performance that was no different than the standard approach. (C) The combined effect of removing the moving average and using basis splines restricts the frequency content of the reconstruction to a three-octave range; for a 500 ms window, this is restricted from 2–16 Hz.

EEG to 16 Hz, so combined with the removal of the moving average, the optimal PCA & spline model constrained the envelope reconstruction's spectrum to a three-octave range (Fig 2C).

## Speech envelope tracking is better than music at all modulation frequencies

Using the number of principal components and splines optimized to the Natural Speech dataset, we computed the reconstruction accuracies for each trial in the current dataset, separately

for each stimulus type, using a leave-one-out procedure (Fig 3A). We analyzed neural tracking of the envelope across a wide range of frequencies by varying the window size of the model between 31.25 ms and 16 s, corresponding to an upper frequency range of 32–256 Hz and a lower range of 0.0625–0.5 Hz respectively. By varying the frequency content of the EEG involved in reconstruction while simultaneously controlling the low frequency cutoff of the envelope, we expected to see an increase in reconstruction accuracy as the bandwidth of the model encompassed the relevant frequency range for envelope tracking for the stimulus (Fig 3B). We can quantify the upper and lower cutoff frequencies of envelope tracking based on the skirts of the reconstruction accuracy plots, which should vary over a three-octave range, corresponding to the range of frequencies where the model's spectrum transitions from not overlapping to fully overlapping the relevant frequency range. However, the reconstruction accuracy needed to be corrected by the variance of the distribution corresponding to chance performance; as the model includes more low frequency content, the variance in the null distribution also increases (Fig 3C). To control for this, we normalized the trial-by-trial reconstruction accuracy with respect to a distribution of null values computed by circularly shifting the envelopes relative to the EEG (Fig 3A and 3C). In this way, reconstruction accuracies were "z-scored" relative to the null distribution.

Overall, we found that all stimuli exhibited above-chance reconstruction accuracies, but the ranges exhibiting this were slightly different. As expected, the trends in the reconstruction accuracy curves increased from chance to near-peak reconstruction accuracy over a three-octave change in the upper cutoff frequency, indicating that EEG tracks the rock envelopes above 2 Hz, vocals and orchestral above 1 Hz, and speech above 0.5 Hz (Fig 3D). Reconstruction accuracies for all stimuli were above chance up to the highest range we could examine for the 512 Hz recorded sampling rate of the EEG. There is considerable evidence of cortical and brainstem tracking of fluctuations at the fundamental frequency of the stimulus pitch [40,41], which might explain the above-chance reconstructions at very high modulation frequencies. Additionally, the ranges of peak reconstruction varied across stimulus types: while the music stimuli had peak reconstructions within 2–32 Hz, speech showed peak reconstruction at a lower frequency range. We thought that the music reconstruction accuracies might be comparable to speech above 2 Hz, but contrary to our expectations, frequency tracking for speech was better throughout the range of frequencies at which performance was above chance (Fig 3E). The difference in reconstruction accuracy between speech and the music stimuli increased with lower frequencies, peaking at the 0.5–4 Hz range (for z-scored reconstruction accuracies for individual subjects, see S3 Fig; for the envelope reconstruction accuracy using original Pearson's r values, see S4 Fig).

We used the Natural Speech dataset to validate that this trend of reconstruction accuracies as a function of modulation frequency was not specific to the data shown here (S5 Fig). The Natural Speech data also showed decreasing reconstruction accuracies for models with frequencies below 0.5 Hz (as in Fig 3D), but z-scored reconstruction accuracies were still above zero at the lowest frequency range we used in this study. This suggests that the limit found in Fig 3D is a consequence of the limited amount of speech data available in this dataset (6–7 trials here compared to 20 trials in Natural Speech), and speech may be weakly tracked at even lower frequencies below our lowest frequency range of 0.0625–0.5 Hz). Note that the Natural Speech dataset was collected using the same EEG system as the current dataset, so the limitation we observed here is unlikely to be an issue with the low-frequency noise floor of the EEG collection.

Because the model hyperparameters were optimized to speech, we considered that this might explain the high reconstruction accuracies for speech. We repeated this analysis using optimal hyperparameters for the music stimuli, but after doing so speech reconstruction still

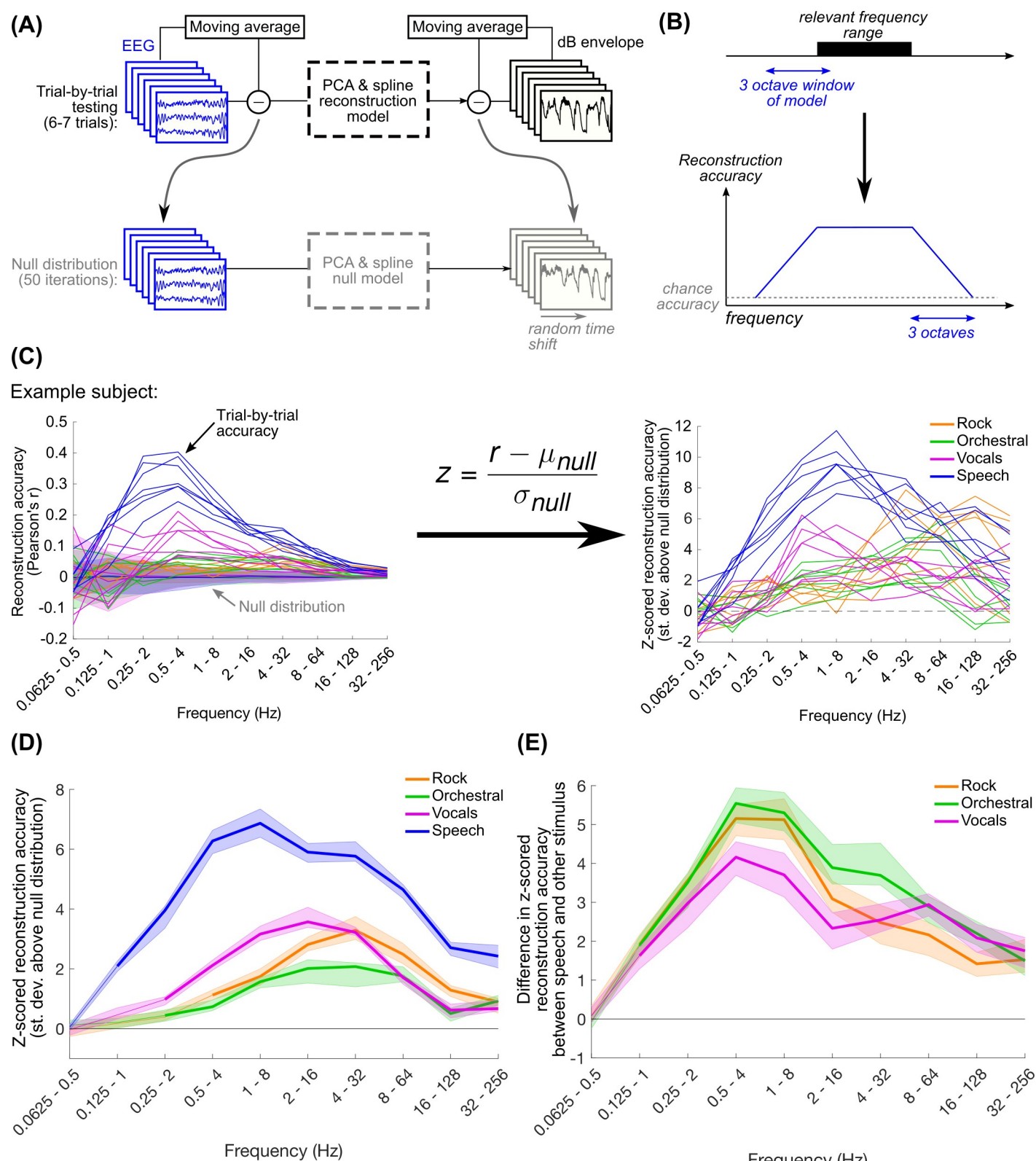

**Fig 3.** (A) For each stimulus type (6–7 trials per subject) the model was iteratively fit to all trials with one left out and tested on the left-out trial. In order to get a null distribution of reconstruction accuracies, we repeated this procedure, leaving out one trial at a time, after randomly circularly shifting the envelopes in each trial by the same amount. This was repeated 50 times for each stimulus type. (B) Schematic of our expectation for how reconstruction accuracy varies with frequency. We

expected that EEG may be tracking a particular frequency range of the stimulus envelope. This could be identified by varying the range of the three-octave model bandwidth. The reconstruction accuracy increases from chance as the model bandwidth overlaps the relevant frequency range, and plateaus when the model bandwidth is fully contained within the relevant frequency range. (C) As lower frequencies are introduced into the stimulus envelope, the variance of the null distribution increases. Because of this, we z-scored the true trial-by-trial reconstruction accuracies relative to the null distribution to ease cross-frequency comparisons. (D) Shown are the median reconstruction accuracies across subjects and trials. Shaded regions show the 95% quantiles of the distribution of 1000 median values calculated using bootstrap resampling with replacement. Thicker lines indicate frequency ranges where median z-scored reconstruction accuracies were significantly greater than zero (one-tailed Wilcoxon signed-rank test with Bonferroni correction for 40 comparisons, p < 0.001). (E) Throughout the frequency range tracked by speech, speech reconstructions were significantly better than all other musical stimuli, with a difference peaking in the 0.5–4 Hz range. Thick lines indicate differences in reconstruction accuracy that are significantly greater than zero (two-tailed permutation test with Bonferroni correction for 40 comparisons, p < 0.001).

outperformed music (S6 Fig). Another potential reason was that the speech envelopes had less variable spectra across trials than the music stimuli (S1B Fig); the speech stimuli came from the same audiobook, but the music stimuli came from different songs with different tempos (see also S1 Table). To control for this possibility, we repeated the envelope reconstruction analysis by fitting and testing models on randomly sampled data within each trial, which maximized the possibility of the testing data containing the same spectrum as the training data (S7A Fig). We found that within-trial reconstruction accuracies were still significantly larger for speech than the other stimuli across all measured frequencies (S7B and S7C Fig).

We also validated if eyeblinks could explain our reconstruction results, perhaps if subjects were more likely to blink at particular events for one stimulus type but not others. The envelope reconstruction analysis was repeated using eyeblink components isolated with independent components analysis (S8 Fig). Using the eyeblink component, reconstruction accuracies were much smaller, and accuracies dropped to zero when only eyeblink peak times were used, implying that eyeblinks could not explain our results.

Together, these results show that envelope reconstruction better captures the neural representation of speech compared to music. The differences in envelope reconstruction performance cannot be explained by spectral variability across trials or by the model optimization procedure. Our results also support the possibility that speech envelope tracking is better due to the relatively larger variability in the speech envelope (Figs 1B and S1).

## Speech envelope tracking at low frequencies is associated with increased weighting of parietal electrodes

Theoretically, the envelope reconstruction model weights spatiotemporal components of the EEG response that most strongly track the stimulus envelope. To understand how these weights correspond to activity evoked by the envelope fluctuations, we inverted the reconstruction models and transformed them from splines and principal components into delays and EEG channels (see Materials and Methods). The EEG response shows temporally consistent peaks and troughs across frequency ranges which vary in magnitude depending on the frequency range represented, indicating an overlap in the spatiotemporal neural activity that is being picked up by each model (Fig 4A).

The temporal variation in the weights is generally consistent across stimulus types, which appears to be particularly true at low frequencies (< 4 Hz) (Fig 4B). Except for increased weighting for speech in channels over posterior scalp, the spatial distribution of the weights was also fairly consistent (Fig 4D). It is also notable that the magnitude of the vocals weights on average are comparable to speech, even though reconstruction accuracies are smaller at these frequencies. At higher frequencies (4–32 Hz), however, the models appear to be of different magnitudes and potentially phase-shifted relative to each other (Fig 4B and 4C; see also S8 Fig for individual model weights for each subject). For example, the speech and orchestral responses are well aligned, but the primary peak for vocals happens about 20 ms earlier and

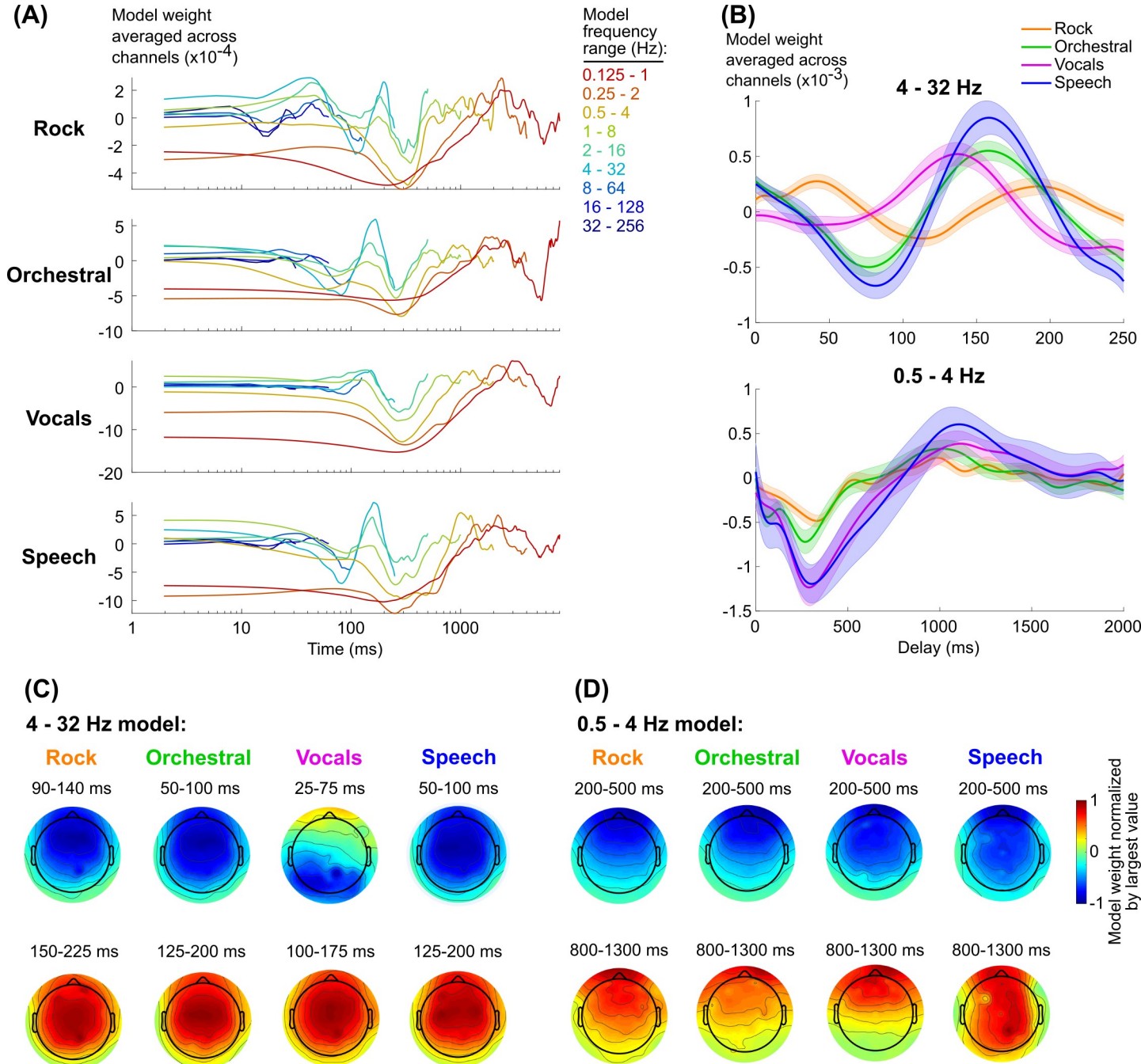

**Fig 4.** (A) Model weights, median across subjects and averaged across channels. Models are color-coded based on their frequency range (see to the right of the plots). The model of the range 0.0625–0.5 Hz was excluded because none of the stimuli exhibited significant neural tracking in this range, and the large values for the weights obscured the trends in the other models. (B) Mean and standard error across subjects of model weights for two frequency ranges (see S9 Fig for the models for individual subjects). (C) and (D) show the topographies of the model weights averaged over the range of delays corresponding to peaks and troughs in the 4–32 Hz and 0.5–4 Hz models respectively. Note that the range of delays vary across stimulus types in the 4–32 Hz model in order to capture similar peaks and troughs.

the secondary peak for rock happens 20 ms later. To what extent do these differences affect neural tracking of the envelopes for the different stimuli? Additionally, speech tracking is better at low frequencies (0.5–4 Hz) than music tracking, but this could be a consequence of

differences in power in the dB envelope at those frequencies (Fig 1B). If so, then a common mechanism would produce comparatively strong neural tracking for speech than music. Is the speech tracking at low frequencies a result of common processing across stimulus types?

To address this question, we reanalyzed the EEG data using envelope reconstruction models trained on all stimulus types and then tested on each trial individually. These stimulus-general models would necessarily capture consistent trends across all stimulus types relevant to envelope tracking. If any trends observed in the stimulus-specific models were involved in tracking the envelope of the respective stimulus, we expected to see higher reconstruction accuracies for the stimulus-specific model than the stimulus-general model. Yet contrary to our expectations for the faster frequencies, speech reconstruction using the stimulus-specific model performed no better than the stimulus-general model, and for all other stimuli the stimulus-specific model performed worse (Wilcoxon signed-rank test with Bonferroni correction for 36 comparisons, p < 0.001) (Fig 5A). In contrast, the stimulus-specific model performed better for speech reconstruction only at 1–8 Hz and 0.5–4 Hz (for reconstruction accuracies for individual subjects, see S3 Fig).

To better understand these differences, we also looked at the difference in performance between the stimulus-specific models and cross-stimulus models trained on one stimulus type but tested on another (S10 Fig). We again found that the model trained and tested on speech did significantly better than the model trained on other stimuli, for both the 4–32 Hz and 0.5–4 Hz models. Notably, we also found that the other same-stimulus models trained and tested on music stimuli seemed to outperform the cross-stimulus models for 4–32 Hz, but not 0.5–4 Hz. This was different from our finding in Fig 5A, where the stimulus-specific models trained on music performed worse than the stimulus-general model. We think this indicates an issue of limited data, since the stimulus-general model was trained on 24–28 trials compared to the 6–7 trials for the cross-stimulus models. However, the improvement in performance produced by the stimulus-general model is still smaller than the difference in performance between speech and the music reconstructions (for reference, see S4 Fig showing the reconstruction accuracy using non-normalized r values). Thus, we do not think that increasing the amount of data would have improved the performance of music envelope reconstructions sufficiently to account for the difference with speech.

We were also concerned that the large frontal negativity at 200–500 ms might be indicative of eyeblinks (Figs 4D and 5C) [42]. We looked at the individual subject topographies and found 4/16 subjects with strong eyeblink-like topographies in their models, but after removing these subjects, only the differences in the 1–8 Hz range were no longer significant, and the topographies continued to show strong frontal activity (S11 Fig).

On average, the stimulus-general models were topographically and temporally similar to the stimulus-specific models (Fig 5B and 5C, compare to Fig 4). To better understand what differences between the speech-specific models and stimulus-general models might have produced the improved reconstruction accuracies for speech, we chose to examine these differences trial by trial. Noting that differences between the stimulus-general and stimulus-specific models were not restricted to specific delays, we instead quantified the scaling and circular shift of the stimulus-general model to best match the stimulus-specific model tested on the same trial (Fig 5D). The circular shift quantifies the change in delay between the two models, and the positively constrained scaling quantifies the change in magnitude.

We used $R^2$ to quantify the fit of the scaled and shifted stimulus-general model to the stimulus-specific model (note that this is different than Pearson's r, which we use as a measure of envelope reconstruction accuracy). The ability to fit the stimulus-specific model varied across stimuli: for both 4–32 Hz and 0.5–4 Hz models, the fit was not significantly better than chance for rock music, and for orchestral the 0.5–4 Hz fit was not above chance (Fig 5E; for reference,

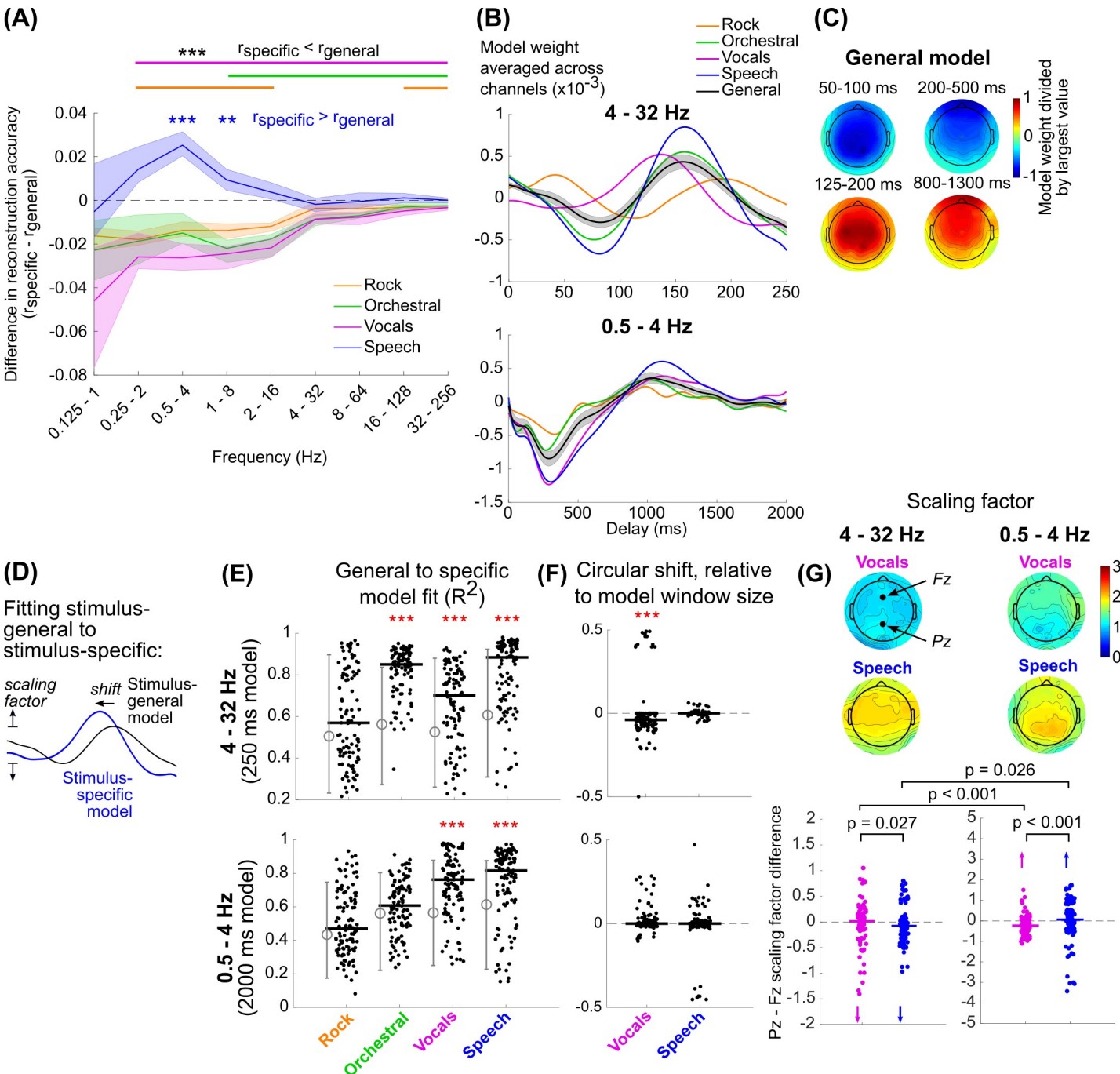

**Fig 5.** (A) The difference between the trial-by-trial reconstruction accuracies using the stimulus-specific and stimulus-general models was then computed. Lines show the median reconstruction accuracy differences across subjects and trials. Shaded regions show the 95-percentile range of bootstrapped resampled median values. Significance values are based on a Wilcoxon signed-rank test with Bonferroni correction for 32 comparisons, *** p < 0.001, ** p < 0.01). The stimulus-specific speech model outperformed the stimulus-general model at 0.5–4 Hz and 1–8 Hz (blue text), while the other stimulus-specific models performed worse than the stimulus-general model for most frequency ranges (black text, solid lines at top). (B, C) The stimulus-general model was quite similar in its temporal (B) and spatial (C) pattern compared to the stimulus-specific models. (D) We assumed that stimulus-general model is a scaled and phase shifted version of each of the stimulus-specific models, so by circularly shifting and scaling the model we could quantify the difference between the model weights. The scaling was computed separately for each EEG channel, but we assumed the shift would be identical for all EEG channels. (E) $R^2$ fits of the scaled and shifted stimulus-general model to each stimulus-specific model on a trial-by-trial basis, based on the summed errors across all EEG channels. Solid black lines show the median values across all trials and subjects. The grey lines to the left of each set of black dots designates the 5% and 95% range of the chance $R^2$ distribution. Red asterisks show the stimulus-types for which the fits were significantly better than chance (Wilcoxon rank-sum, p < 0.001). (F) Distribution of circular shifts plotted identically to E. Red asterisks show distributions whose medians are significantly different than zero (Wilcoxon signed-rank, p < 0.001). (G) Above, topography of scaling factors for vocals and speech. The scaling factors at channels Fz and Pz were

then compared for each trial. Below, individual scaling factor differences (Pz–Fz) for each trial and subject for vocals (magenta) and speech (blue). Arrows in each plot indicate individual points that were outside of the y-axis limits in the plot. Comparisons between vocals and speech for each frequency range are based on a Wilcoxon rank-sum test. Comparisons between frequency ranges are based on a signed-rank test.

see also the $R^2$ values of the stimulus-general to stimulus-specific model fits without scaling or circular shifting in S12 Fig). This indicates that there were either differences in the rock and orchestral stimulus-specific models compared to the stimulus-general model, or that the rock and orchestral models were too variable to be properly fit by the stimulus-general model. The vocals and speech models were well-fit by the scaled and shifted stimulus-general model. Additionally, the circular shift of the stimulus-general model matched the expected shift from the averaged model for vocals (Fig 5F), showing a slight negative shift that was equal to amount of lead in the peaks of the vocals model relative to the stimulus general model.

We also looked at the scaling of the stimulus-general model as a function of EEG channel (Fig 5G). Interestingly, the topographies of the scaling factors differed slightly for vocals and speech. For 4–32 Hz, the scaling was largest in frontal and central channels, consistent with the spatial patterns of the model weights for all other stimuli (Fig 5C), and consistent with the weights found for models constrained to this frequency range in prior work [19,43]. For 0.5–4 Hz, the scaling was largest in parietal channels, where model weights were somewhat larger for speech than the other stimuli (Fig 4D).

To test these topographic differences statistically, we looked at the difference between parietal channel Pz and frontal channel Fz (Fig 5G). For the 4–32 Hz model, the difference between Pz and Fz was significantly larger for vocals than speech, although this effect was weak (Wilcoxon rank-sum test: z = 2.22, p = 0.027). For the 0.5–4 Hz model, this relationship flipped and speech showed a more positive Pz-Fz difference (z = -4.20, p < 0.001). Comparing between frequency ranges, vocals showed a more negative Pz-Fz difference for low frequencies than high frequencies (Wilcoxon signed-rank test: z = 4.02, p < 0.001). Again, this relationship was flipped for speech, though the effects were weaker (z = -2.22, p = 0.026).

Taken together, we found that, at higher frequencies, temporal responses seem to differ between stimulus types, but these differences are not consistent enough or robust enough to outperform a model fit to all stimulus types. However, the stimulus-specific speech model outperformed the stimulus-general model at lower frequencies. Topographic differences for speech were weak, but they appeared to show increased weighting over parietal channels, opposite the increased weighting over frontal channels for vocals.

## Drums in rock music tracked at multiples of the musical beat frequency, but tracking was worse than speech

Our previous analyses demonstrated that EEG tracks both speech and music above 1 Hz, but speech is tracked more strongly than the music stimuli (Fig 3D and 3E). To some extent, this could be explained by the greater magnitude of fluctuations in the dB envelope for speech, which is a consequence of the sparsity of speech [44,45]. In contrast, the rock and orchestral stimuli were multi-instrumental, which have reduced envelope power and reduced sparsity. However, it is plausible that the brain might more strongly track one of the instruments in these stimuli, perhaps matching the strength of tracking observed for speech. Furthermore, the rock reconstruction accuracies were not significantly different than zero using models with frequency content strictly below 2 Hz. This contradicts some evidence showing neural tracking of music with note rates below this frequency [7,46] (note also that 3 out of 10 of the rock songs in this study had tempos, or musical beat rates, below 2 Hz). Perhaps individual instruments may be tracked better at lower frequencies.

To address these possibilities, we re-examined EEG neural tracking of the rock stimuli by quantifying reconstruction accuracies to the envelopes of individual instruments. This was possible because the rock songs were multi-tracked, so to retrieve the envelope of an individual instrument all of the other instruments in the track were muted. We focused our analysis on the vocals, guitar, bass, and drums because these instruments were present in all songs. Models were then trained and tested on individual instruments using the EEG data collected while subjects were listening to the multi-instrument rock songs.

Compared to all other instruments, the drums were reconstructed best; only this instrument produced z-scored reconstruction accuracies that were not significantly lower than the accuracies for the multi-instrument rock envelope (Fig 6A and 6B; a Wilcoxon signed-rank test with Bonferroni correction for 40 comparisons showed that the drum z-scored reconstruction accuracy was slightly better than the accuracy for the full rock envelope for the 8–64 Hz model: z = 3.28, p = 0.042) (see S13 Fig for the reconstruction accuracy curves and differences for individual subjects). However, the z-scored reconstruction accuracy was still considerably less than what we observed for speech (Fig 3D).

At the start of this study, we assumed that EEG might track the amplitude fluctuations in music as a consequence of tracking rhythmic structure [35,47]. To better understand the neural tracking of the drums, which strongly suggests tracking of musical rhythm, we looked at the power spectral density of the drums reconstructions for each track averaged across subjects. We focused particularly on reconstructions using the 2–16 Hz, 4–32 Hz, and 8–64 Hz models which produced the best reconstructions on average (Fig 6C). We then subtracted the spectra from null spectra generated by randomizing the phases in the reconstructions for each subject and averaging these randomized reconstructions (see [48,49]) (Fig 6D). From these adjusted spectral values we identified the peaks occurring at the tempo of the music (see Materials and Methods) as well as 2 – 4x the tempo, since the peak energy may occur at multiples of the expected musical beat frequency of the music based on the acoustics [50] or neural activity following subcortical processing [51].

Firstly, we found that, most often, the peak energy in the reconstruction often did not occur at the tempo of the music (the left-most dots of Fig 6E). Interestingly, the peak energy for most reconstructions was restricted to a 2–8 Hz range, peaking around 5 Hz. This result supports our earlier observation that reconstruction accuracies for the rock music and drums were best using models fully encompassing this frequency range (Figs 3D and 6A). The peak energy around 5 Hz also supports other observations of frequency tracking to real music [35] as well as rhythmic synthetic stimuli [47,52]. The reduced energy below 2 Hz can be explained by model constraints, which were restricted to a low cutoff of 2 Hz. However, all models contained energy above 8 Hz, suggesting that the upper cutoff reflects neural tracking limits and not the model specifications. This could be a consequence of the evoked responses to the musical events, which may have little energy above 8 Hz.

Overall, EEG tracks drums better than any other instruments at a rate associated with the rhythmic structure of the music, but not necessarily at the musical beat. In particular, this tracking peaked around 5 Hz and tracking was not observed above 8 Hz. However, the neural tracking of the drums alone was still worse than what we observed for speech, implying that the difference in reconstruction accuracies that we observed between the speech and music could not be explained by passive neural tracking of, or potentially attention to, individual auditory objects in the mixture.

## Discussion

In this study, subjects passively listened to separate trials containing speech, rock music, orchestral music, or the vocals from the rock music. Using the recorded EEG on each trial we

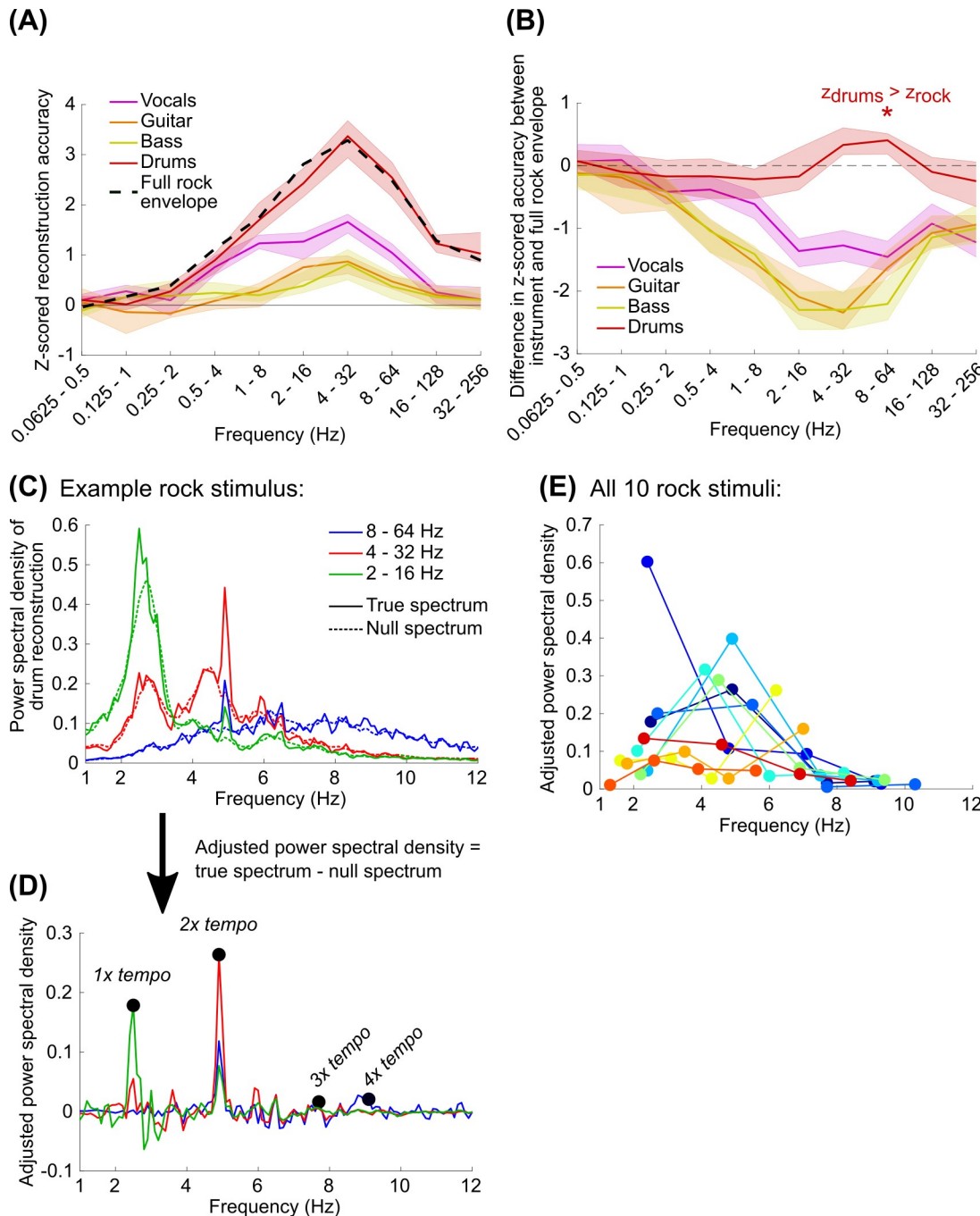

**Fig 6.** (A) Using the EEG data recorded while subjects were listening to the rock songs, we trained and tested PCA & spline models on the dB envelopes for the vocals, guitar, bass, and drums individually. Z-scored reconstruction accuracies were quantified as in Fig 3A–3D. All instrument envelopes were reconstructed above chance when the model included frequencies above 2 Hz (Wilcoxon signed-rank test: p < 0.001 with Bonferroni correction for 40 comparisons). The full rock envelope, shown with a dashed black line, is equivalent to the values shown in Fig 3D. (B) Pairwise differences between the z-scored reconstruction accuracy for the full envelope and the envelope for each individual instrument. The z-scored reconstruction accuracies for drum were not significantly different than the same pairwise reconstruction accuracies for the rock envelope based on the multi-tracked recording with all instruments, except for the 8–64 Hz model where reconstruction accuracies were slightly but significantly better than full rock envelope (Wilcoxon signed-rank test with Bonferroni correction for 40 comparisons: z = 3.28, p = 0.042). (C) Welch's power spectral density of the reconstructions was computed for each stimulus and averaged across subjects. The noise floor of the power spectra is shown with dashed lines. (D) We then adjusted the power spectral density by subtracting the true spectrum from the average of the null spectra, which made the peaks associated with temporally coherent

reconstructions across subjects clearer. The maximum values in the adjusted power spectral density were then identified relative to the expected tempo of the music (1x tempo) as well as 2x to 4x the tempo. (E) Each of the 10 rock stimuli are plotted as a different color, and each dot corresponds to 1 – 4x the music's tempo with increasing frequency. The darkest blue line and dots correspond to the example stimulus shown in C and D.

used linear modeling with PCA to control spatial correlations in the EEG and a basis spline transformation to control temporal correlations. These transformations, in addition to high-pass filtering, restricted the frequency content in the EEG used for envelope reconstruction. We then compared reconstruction accuracies for a wide range of frequency bands in order to identify frequency ranges that were tracked best by each stimulus type. We found, however, that speech was tracked better than the other stimuli for all frequencies we examined, and speech tracking continued below 1 Hz where all other stimuli showed reconstruction accuracies not significantly better than chance. Closer inspection of EEG tracking for the rock music showed that the drums, or perhaps more generally the rhythmic structure, was tracked most strongly around 5 Hz, but this tracking was still worse than what was observed for speech in this frequency range. For modulation frequencies above 1 Hz, a model trained on all stimuli did just as well at reconstructing speech and better for the music stimuli, suggesting common mechanisms involved in this range. However, stronger tracking of speech below 1 Hz appeared to be associated with increased weighting of parietal channels. Together, this suggests that both music and speech are tracked above 1 Hz using mechanisms with largely similar spatio-temporal responses in the EEG, but below 1 Hz mechanisms exist that more strongly track speech than music.

Our aim in this analysis was to understand how the acoustics of speech and music might produce observed differences in envelope reconstruction accuracy. In the average spectrum, speech shows a second, lower frequency peak around 0.3 Hz, within the range where we observed neural tracking that was not present for the other stimuli (Figs 1B and S1). This energy corresponds to phrase-level fluctuations in the envelope. Thus, it is possible that the neural tracking we observed for speech is due to the relatively high energy in that frequency range compared to the other stimuli. However, it is surprising that the enhanced neural tracking was not observed for the vocals, which also showed fluctuations in a slightly lower frequency range (~ 0.2 Hz). Unlike the rock instruments, whose envelope reconstructions also did not match speech performance, the vocals stimuli were presented in isolation, suggesting that attention to an individual auditory object also may not explain the relatively high performance observed for speech. Additionally, we found that the models for vocals did not have enhanced weighting over parietal channels, like we observed for speech (Fig 5G). The importance of low-frequency tracking (within the range generally considered "delta" in most studies) for speech is not new [14,29,53], but to our knowledge no study has suggested that low-frequency tracking is stronger for speech than other naturalistic stimuli. Recent studies also found that parietal weighting was increased for tracking phoneme and word surprisal [54] as well as semantic tracking of speech for native speakers but not non-native speakers [55]. The parietal weighting could be indicative of language-specific processing in the posterior temporal lobe [56] which was recently shown to be absent when listening to music [57], but because the activations are broad and without source localization it is difficult to identify the location definitively. Still, there is some evidence that low-frequency tracking of speech may be produced by a domain-general mechanism [58]. A future study could determine if the low-frequency tracking we observed here is specific to speech by presenting both speech and single-instrument stimuli with phrase-level fluctuations in amplitude. Such a study could also clarify if the increased parietal weighting was not observed for vocals simply because there was not

enough modulation power in their envelopes. If the low frequency tracking truly is unique to speech, we would expect that other sounds explicitly designed to have high energy in that frequency range will not elicit the same level of neural tracking as observed for speech.

Additionally, we found that speech reconstructions were better than the various musical stimuli across all frequency ranges. Our specific observation that vocals reconstructions were overall worse than speech seems to contradict results based on phase coherence showing increased tracking of music than speech [59,60]. One concern with our results is that the music envelopes, particularly rock and vocals, are more periodic than speech, and mismatches between their autocorrelation functions could hinder envelope reconstruction. If so, we would expect within-trial music reconstructions to perform better, since the autocorrelations are more likely to be the same for testing data in the same song than in a different song. But that was not the case (S7 Fig).

Note, however, that we cannot rule out that the larger modulation power for speech produces better reconstruction accuracies, particularly at frequencies corresponding to the theta range (4–8 Hz) (S1 Fig). Even though we normalize the variance of the stimulus envelopes and the EEG before model fitting and testing, larger modulations in the original stimulus could improve the signal-to-noise ratio of the EEG signal tracking the envelope (Fig 1B). Relatedly, the magnitude of the weights for the speech models suggests that the evoked responses to speech are larger, which increases the signal-to-noise ratio in the EEG and could produce greater reconstruction accuracies of the speech envelope. This reasoning does not entirely match the results for music, though, because rock music produced reconstruction accuracies on par with the vocals alone even though the average spectrum was smallest of the three stimulus types. The best way to address this concern would be with a follow up experiment that includes speech and music stimuli that are better matched in modulation power. Alternatively, more regions of auditory cortex are recruited when listening to speech than music [61,62], which could also explain the increased EEG activation we observed for speech. Furthermore, some studies have argued that temporal modulations are more important for speech processing, while spectral modulations are more important for music ([63,64]; but see [65]). It is possible that a different acoustic feature might be tracked better by the EEG than the temporal envelope during music. Lastly, many studies of EEG processing of speech have shown the importance of speech features in affecting evoked responses in this frequency range, including phoneme coding [19], phoneme probability encoding [66], surprise and uncertainty [14,29].

Our analysis focused on isolating specific frequency contributions to envelope reconstruction while also identifying the spatiotemporal characteristics of the EEG responses. One intention of the analysis in this study was to bridge the gap between evoked and oscillations-based interpretations of the neural tracking of speech and music. Here our stimuli are naturalistic sounds, but we use frequency-constrained modeling to isolate the spatiotemporal responses which track amplitude modulations in these sounds, showing increased tracking at frequencies commonly associated with theorized oscillatory tracking for parsing sounds (theta: 4–8 Hz, delta: < 4 Hz). Given our results, it seems reasonable that theories of evoked responses and phase-related tracking can be parsimoniously explained by frequency-tuned evoked responses (see also [27]). Further modeling work comparing the theories of evoked tracking and oscillations-based tracking will also be beneficial to reconciling their differences and the observations in EEG (for example, [46]).

When reconstructing the envelopes of individual instruments in the rock music, we found that drums were reconstructed best, and reconstructions showed that peak energy was usually between 2–8 Hz with a maximum around 5 Hz. This analysis was inspired by several studies that have examined the frequency content of EEG or MEG in order to quantify the neural tracking of musical beats [35,47,52,67]. We did not find strong or consistent tracking at the

frequency of the tempo, but our results suggest a particular importance for neural tracking around 5 Hz, consistent with a recent study of EEG tracking of Indian music [35]. Other studies have found significant neural tracking at the meter of the music [16,22], and it is possible that our approach of examining envelope reconstruction with linear modeling fails to capture neural tracking at this frequency range, especially if it is only weakly present in the original envelope. The potential importance of neural tracking at a multiple of the beat frequency has been observed in other studies modeling stages of subcortical processing [50,51] but they have not identified the 2–8 Hz frequency range as specifically important. Interestingly, this result is also contrary to an analysis of temporal modulations in speech and music, which showed a prominence of 5 Hz for speech and 2 Hz for music [50]. 5–8 Hz encompasses the limit of auditory-motor synchronization found across various studies [68]. However, these fast events might be relevant for representing the smallest temporal unit in a musical piece defining the grid on which the musical beat, rhythmic structure, and meter are based [69]. Understanding the relationship between peak synchronization to musical events, the limits of synchronization, and listener experience could be relevant for biophysical modeling of music perception in the future.

Quantifying the information carried by EEG for decoding speech is an active area of research in brain-computer interfaces and auditory attention decoding, which is now focused primarily on using reconstruction accuracy to identify a subject's locus of attention [11,70–72]. Our interest here was also to use reconstruction accuracy as a means of quantifying how sufficiently the spatiotemporal responses represent neural tracking of the stimulus type. But we think that our observation of stronger low-frequency tracking ($< 1$ Hz) for speech is notable for auditory attention decoding work. Low-frequency tracking may relate to several cognitive aspects of speech processing such as semantics, prosody, surprise, attention, comprehension, and language proficiency [14,29,36,55,73,74]. If other naturalistic sounds are not sensitive to this frequency range, then there could be considerable benefit to focusing on this frequency range to identify the locus of attention of a talker and isolate the most relevant speech feature to which a user is engaged.

## Materials & methods

### Ethics statement

The experimental procedures were approved by the Ethics Committee for the School of Psychology at Trinity College Dublin, and all subjects provided written consent at the beginning of the experiment.

### Experiment and EEG recording

Stimuli consisted of seven approximately three-minute segments from an audiobook in English ("speech" stimuli), ten rock songs including vocals each 3.5–5 minutes long, ten segments of orchestral pieces that were 3–4.5 minutes long, and ten tracks of the vocals from the rock songs (2.5–4.5 minutes) (see S1 Table for a list of the stimuli and more detailed information). All of the rock songs were originally multitracked, so the other instruments were muted in order to isolate the vocal track for the vocals stimuli, and silences were manually shortened to reduce the overall length of the track. Each stimulus was preceded by a 10 ms voltage click occurring 0.5 s before the start of the stimulus; the click triggered an Arduino to provide a code to the EEG data collection system signifying the start of the audio in the recorded EEG data to microsecond precision.

17 subjects took part in the experiment. One subject's data had issues with trigger timing that produced shorter EEG recordings than the actual stimulus durations, and their data was

excluded from further analysis. In total, data from 16 subjects (7 female; ages 18–44, median 22) were included. Subjects listened to the stimuli at a comfortable sound level via Sennheiser HD650 headphones. Stimuli were presented to the subject using Presentation software (Neurobehavioral Systems). 128-channel EEG and two mastoid channels were recorded (Biosemi ActiveTwo) at 512 Hz as subjects passively listened to each of the sounds in the following order within each block: rock, orchestral, vocals, speech. An additional track consisting of tone pips was also included and was presented every 4–5 trials, but the data was not analyzed for this study. Each subject listened to 6–7 trials of each stimulus type.

After the session, the EEG channels were referenced to the average of the mastoid channels. No other preprocessing was applied to the EEG data prior to modeling.

## Extracting the stimulus envelope

To get the stimulus envelopes, the stimulus waveform was filtered with a bank of 32 gamma-chirp filters [75] logarithmically spaced between 100 Hz and 8 kHz. The amplitude of the Hilbert transform of each channel was then averaged across frequency and normalized to have a peak amplitude of 1 V. Prior to converting the envelope to dB, in order to prevent discontinuities at zero values in the envelope, all voltages below $10^{-5}$ V (equivalent to -100 dB V) were set to $10^{-5}$. The dB envelope was then resampled to 512 Hz for the speech and music data. We chose to work with the dB envelope because it is more linearly related to the perception of loudness and the EEG response to sound level than the raw voltage values [76–78]. Additionally, in contrast with the original voltages that are strictly positive, we thought assumptions of Gaussian distributed errors for linear modeling might be more appropriate for the dB envelope.

We also wanted to visualize the power spectra of the envelopes to compare them across stimulus types. However, the original power spectra have a 1/f slope typical of natural signals [79] which makes the peaks in the spectra and comparisons between stimulus types difficult to see. A standard approach is to remove the 1/f slope with linear regression in log-frequency, but this reduced the apparent magnitude of the peak in the speech envelope energy below 1 Hz, and we thought that seeing the absence of energy below 1 Hz considering the strong speech envelope tracking we observed in that range would be misleading to the reader.

Instead, for Fig 1B, we compute the ratio of the envelope power spectra relative to the average EEG power spectra, in order to quantify the hypothesis that EEG tracking of the envelope is a direct replicate of the envelope itself and that the differences in reconstruction accuracy are due to differences in envelope variance at each frequency range. Specifically, each dB envelope was zero-centered, and a 16 s moving average was subtracted (equal to the maximum model delay, see "Quantifying cross-frequency model performance" in the Materials and Methods). All envelopes were then normalized by the square root of the average variance across all stimuli. Then, the power spectra of individual envelopes were computed. To compute "EEG noise", the EEG data in each trial was averaged across channels, the 16 s moving average was subtracted, and the averaged EEG was z-scored. Then, the power spectrum was computed on the z-scored EEG data. Power spectra of the envelopes were averaged across trials (including all stimulus types) and subjects, and the ratio of the envelope power to EEG noise power was computed, where the denominator of the ratio was using the trial- and subject-averaged EEG spectrum.

The Envelope power to EEG power ratio is theoretically proportional to the signal-to-noise ratio of the neural response to the envelope, assuming that the neural response is a scaled version of the dB envelope (the null hypothesis) and the EEG noise is proportional to the average EEG spectrum. Note that while we have no knowledge of the linear relationship between dB

envelope and the neural response, a change in the scaling would multiply these ratios for each stimulus type identically and maintain their relationships to each other.

To see the envelope power spectra prior to this normalization procedure, see S1 Fig.

## Effects of frequency content on regularization-based modeling of envelope tracking

Our initial goal to compare speech and music neural tracking raised several issues associated with stimulus differences. Firstly, the shapes and spectra of the envelopes are very different across stimulus types (Fig 1), so by comparing envelope reconstructions to the broadband envelope, certain frequency ranges may contribute more to the overall error; if a stimulus type isn't being tracked at low frequencies, it might produce a worse reconstruction accuracy despite reasonable tracking at higher frequencies. Secondly, the speech stimuli had much less cross-trial variability than the other stimulus types, so a trial-by-trial approach to modeling the envelope would be poorer for the music stimuli than for the speech stimuli. Because the rhythms of music are known to be more varied than speech [50], we think this issue is unlikely to have changed using a different stimulus set using the same stimulus types.

Typically, envelope reconstruction model weights are constrained using ridge regression or some other form of regularization which minimizes the variance of the weights [25,38,80]. Ridge-type regularization addresses issues of multicollinearity that are present both spatially (between neighboring EEG channels) and temporally by down-weighting the contribution of low-variance principal components in the input, effectively acting like a low-pass filter [81,82]. For the purposes of comparing neural tracking for stimulus types with different spectral characteristics, this is problematic because of the lack of control over the frequency effects of regularization; while we could show that reconstruction accuracies are different between different stimulus types, we would be unable to claim that these differences are not due to spectral differences in the envelopes.

Additionally, depending upon the range of delays used, model weights will affect the spectral content at higher frequencies, but leave lower frequencies untouched. To simulate this, a regularized reconstruction model fitted to the speech stimuli using EEG delays between 0 and 500 ms (a model width of 500 ms, S2A Fig) was applied to a broadband noise input with a flat spectrum. The output of the model shows variation in the spectrum above 1 Hz, but little effect on the magnitudes of the spectrum below 1 Hz (S2B Fig). As a consequence, low-frequency neural tracking may be present and factor into the reconstruction accuracy measures. To demonstrate this in S2C Fig, we reconstructed the different stimuli using models trained on each stimulus type separately, first using EEG and stimulus envelopes that were highpass filtered above 0.1 Hz by subtracting the moving average of a 10 s window, and then using a highpass filtered envelope and EEG after similarly removing energy below 2 Hz. Speech reconstruction accuracies significantly decrease when frequencies below 2 Hz are removed (Wilcoxon signed-rank test with Bonferroni correction for 4 comparisons: $z = -7.2$, $p < 0.001$), while the reconstruction accuracies for the music stimuli significantly increase when the low-frequency content is removed (rock: $z = 6.6$, orchestral: $z = 4.0$, vocals: $z = 6.2$; $p < 0.001$ for all stimuli).

## Envelope reconstruction with PCA & spline transformation

All code used to create and test these models is available on gitlab (https://gitlab.com/eegtracking/speech_music_envelope_tracking).

The model we used for this study was a linear model that reconstructed the stimulus dB envelope using the principal components of the EEG and a spline basis to focus on lower frequencies. The PCA and spline transformations control multicollinearity in the EEG data.

Transforming the EEG channels into principal components reduces spatial correlations between channels; this was done using Matlab's built-in *pca* function. The basis spline transformation reduces the delayed EEG values to a lower-frequency representation using cubic basis splines. Specifically, cubic splines are piecewise polynomial functions defined by a sequence of knots such that the first and second derivatives of the function are continuous at each knot. For a sequence of knots, cubic splines can be collated across the dependent dimension of a function, and any cubic spline function defined by those knots can be represented by linear combinations of these basis splines. The collation matrix was constructed using functions from Matlab's Curve Fitting Toolbox: *augknt* to construct the sequence of evenly-spaced knots along the range of delays, and *spcol* to create the basis spline matrix. The number of splines is defined by the number of spline knots, so the spline basis can also be defined by the sampling rate of the knots (this designation is used in Fig 2B).

When fitting the model, first the moving average of the stimulus envelope and the EEG, using a window size equal to the maximum lag in the model, was subtracted. One trial was left out for testing, and the rest of the trials were used for model training. Next, the EEG data in the training trials were converted into principal components, and the principal components were z-scored for each trial. To create the design matrix for model fitting, each principal component was delayed from 0 ms delay up to the maximum delay in the model. For a lag matrix for a principal component $X_d$, the lags were converted into a matrix of basis splines $X_s$:

$$X_s = X_d S (S^T S)^{-1}$$

where S is the collocation matrix of basis splines with delays along rows and basis cubic splines along columns. Both the spline-transformed design matrix and the stimulus envelope were z-scored for each trial, and then concatenated across trials for model fitting. The model was then computed using linear regression:

$$w_s = (X_s^T X_s)^{-1} X_s^T s(t)$$

where *s(t)* is the stimulus envelope. The first and last 16 seconds of each trial were left-out of model fitting and testing in order to avoid potential edge artifacts produced by the removal of the moving average using the lowest-frequency model (the model with the largest window) in subsequent modeling (see Fig 3). This included the click and 0.5 s of silence before the stimulus started, so overall the first 15.5 s and the last 16 s were left-out of each trial.

For testing, the EEG data in the left-out trial were transformed into principal components using the same transformation matrix calculated for the training data. The delayed principal components were transformed into a spline basis as described for the training data and the stimulus envelope for the testing trial was reconstructed by $s'_{test}(t) = X_{s,test} w_s$. Throughout, reconstruction accuracy was quantified based on the Pearson's correlation between the original stimulus envelope and the reconstructed stimulus envelope.

For later analysis of the model, model weights were converted from splines to delays for each principal component by $w_d = S w_s$.

## PCA & spline model optimization

The envelope reconstruction model contained three model hyperparameters: the number of principal components to use, the window size of the model, and the number of basis cubic splines (or, equivalently, the sampling frequency of the spline knots). The model window size and the number of splines constrain the lower and upper cutoff frequencies contained within the model, respectively. As such, we chose to optimize the number of splines with respect to the window size, with the intention of manipulating both in tandem when examining neural

tracking across frequency bands. Our method of identifying the optimal hyperparameters is similar to other modeling approaches that identify the hyperparameters using a cross-validation approach [61,62].

We used the Natural Speech dataset to identify the optimal model parameters [36]. In this dataset, subjects listened to an audiobook for 20 trials each approximately 180 s long (a subset of these segments was used for the speech stimuli in the current study). The dataset contains the raw EEG and the stimulus envelope at 128 Hz. We first converted the stimulus envelope into dB; we did not threshold at -100 dB V like we do later for the envelopes in the current study (see "Extracting the stimulus envelope" above) and instead used the real value of the $\log_{10}$ of the envelope, since thresholding failed to reduce the effect of low-amplitude artifacts in the envelope and the real component of the log value seemed to be more robust to these issues. The EEG was referenced to the average of the mastoid channels.

For model optimization, we used a 500 ms window because of its common use in envelope reconstruction [13,37–39]. We first removed the moving average of a 500 ms window from both the dB stimulus envelope and the EEG. Then we transformed the EEG into principal components and fitted a model using basis splines. We compared the performance of this model to one based on ridge regression of the original EEG using regularization parameters between 0 (no regularization) and $10^8$; most often the optimal regularization parameter was around $10^4$ to $10^5$. PCA & spline models were optimized relative to the performance of the regularized model using a grid search, using 8, 16, 32, 64, or 128 principal components, and 7, 11, 19, 35 splines (corresponding to a spline knot sampling frequency of 8, 16, 32, 64 Hz respectively) or all 64 lag parameters of the model without a spline transform. Performance was evaluated on a left-out trial and trained on the other 19 trials.

To estimate the frequency content of the resulting optimum model (Fig 2C), we used broadband noise as the input to the model. First, the moving average of the noise using a 500 ms window was removed. Next, the design matrix was created from the noise using delays -250 to 250 ms, and the matrix was converted into 19 basis splines (32 Hz sampling frequency of the knots); we set 0 ms as the center delay because when 0 ms was the first or last delay (for example, 0–500 ms), the edge spline was most heavily weighted and produced edge affects in the resulting spectrum, which inappropriately represented the low-pass effects of the basis splines generally. We then used linear regression to estimate the best fit between the spline-transformed design matrix and the original noise with the moving average removed, which we used to reconstruct the noise. The filter resulting from the combined moving average removal and the spline transformation was computed by getting the ratio of the reconstructed noise to the original broadband noise. Fig 2C shows this ratio in dB V.

## Quantifying cross-frequency model performance

We varied the frequency content of the model by changing the window size while retaining the same number of splines (19 splines), which maintained a 3-octave bandwidth. The low frequency cutoff of each range corresponds to the reciprocal of the maximum delay in the model, so by examining models with maximum delays from 31.25 ms to 16 s, we varied the frequency content of each model from a 32–256 Hz range to a 0.0625–0.5 Hz range respectively. Reconstruction accuracy was quantified based on the Pearson's correlation between the envelope reconstruction of the left-out trial and the original envelope.

Because chance performance varies with the frequency content of the stimulus envelope and EEG (Fig 3C), we calculated a null distribution for each frequency range separately. For each iteration, the stimulus envelopes were randomly circularly-shifted and the same shift was applied to all trials. Then, one of the trials was randomly chosen and left out for testing, and

the rest of the model fitting and testing procedures were applied as described above. This was repeated 50 times to get a distribution of null accuracies for each stimulus type and for each subject. The true reconstruction accuracies were normalized relative to the null distribution and reported in standard deviations relative to the null distribution of accuracies to get the "z-scored" reconstruction accuracy (see Fig 3C).

## Getting a spatiotemporal EEG response from reconstruction model weights

Next, we wanted to understand how neural responses were represented by the reconstruction models. However, reconstruction models, also known as "backwards" models, are not interpretable without accounting for autocorrelations in the data, since the resulting reconstruction model could produce weights which cancel out irrelevant autocorrelations and do not necessarily represent a neural response [83]. To account for this possibility, we first inverted the model, using principal components and a spline basis, into a "forward" model that is a better representation of the spatiotemporal evoked neural response [43,80]. The transformation was a modification of the approach used by [83]. The forward model $a_S$ was calculated from the backward model $w_S$ by:

$$a_S = \frac{1}{N}\left(X_S^T X_S\right)w_S$$

where $N$ is the total number of sample points in the training data and $X_S$ is the design matrix for the training data. In contrast with the original approach by [83], this scales the weights assuming that the input is a representation of the stimulus envelope with unit variance. We chose to use this approach to make the forward models more comparable between stimulus types. Then, the "forward" model weights were converted into delay weights (see Materials and Methods: "Envelope reconstruction with PCA & spline transformation").

The model was then converted from principal components into EEG channels. Because the principal components were z-scored prior to fitting the model, each principal component was then multiplied by the ratio of the variance of the principal component to the summed variance across 128 principal components. Firstly, this ensures that the scaling of each principal component matches its original contribution to the EEG signal. Secondly, the EEG variance can increase considerably at lower frequencies, so the variance in the principal components tends to be larger for the lower frequency EEG range. Dividing by the total variance across components normalizes this effect of frequency on EEG variance and allows for cross-frequency model comparison.

## Fitting the stimulus-general model to the stimulus-specific model

We wanted to understand how differences between the stimulus-specific models might correspond to differences in reconstruction accuracy. We addressed this by looking at the difference in performance of a stimulus-general model, fit to all stimulus types, relative to the stimulus-specific model. For the stimulus-general model, trials from all stimulus types (23–27 trials) were included in training data, and the series of steps for computing the principal components, normalizing the components, and creating the basis spline design matrix were identical to those described in the Methods: "Envelope reconstruction with PCA & spline transformation". Importantly, both the stimulus-general and stimulus-specific models were tested trial-by-trial, so we used the difference in reconstruction accuracies (Pearson's r) for the two models on each trial to quantify the change in model performance.

We then wanted to systematically quantify the changes in weights between the two types of models. In a linear model, the effects of model weights are combined to optimize the

reconstruction, which can complicate interpretation [84]. Thus, we thought it unlikely that differences at specific delays would be sufficient to explain the change in reconstruction accuracy. Subjective examinations of the difference in weights showed that the models differed primarily in the amplitude ("scaling factor") and delay ("shift") relative to each other, even though the shapes were similar. Thus, we thought it would be most appropriate to examine the relative circular shift and scaling of the stimulus-general model that would match the stimulus-specific model at each channel.

To compute the circular shifts and scalings, both the stimulus-specific and stimulus-general models were centered to have a mean weight of zero for each channel. Next, for each circular shift, the best scaling factor for each EEG channel was computed using linear regression. Then the $R^2$ model fit across all channels was computed. The shift producing the maximum $R^2$ was identified as the optimum model for that trial. All analyses were then based on the model fits ($R^2$), circular shifts, and scaling factors computed on a trial-by-trial basis. Note that the $R^2$ values were considerably lower without any shifting or scaling of the stimulus-general model, showing that this procedure indeed improved the fit to the stimulus-specific model weights (S12 Fig).

Additionally, to evaluate the goodness of fit, we computed a null distribution of $R^2$ values by randomizing the phases of the stimulus-specific model and refitting the stimulus-general model as described above. This procedure ensured that the spectral amplitudes of the stimulus-specific model would remain the same, since the high $R^2$ values could be due to relatively high signal power at low frequencies, while ensuring that the temporal relationship between frequencies in the model was destroyed. 20 null $R^2$ values were computed for each trial, and the distribution of all null values was combined across trials and subjects for statistical testing relative to the true distribution of $R^2$ values (see Fig 5E).

## Quantifying power spectra of drum reconstructions relative to the music's tempo

While we found higher reconstruction accuracies for speech than the other music stimuli, we considered the possibility that EEG may track individual instruments in the rock or orchestral music more so than what we observed using the envelopes calculated from the multi-instrument mixture. To examine this further, we computed instrument-specific models using the EEG data recorded during the presentations of rock stimuli by training and testing on the envelopes for the vocals, guitar, bass, and drums in the rock tracks. Because the rock songs were multi-tracked, we muted all other instruments in order to get the waveform for the individual instrument and computed the envelope as described previously (Methods: "Extracting the stimulus envelope").

After we found that drums exhibited the best reconstruction accuracy of all of the rock instruments (Fig 6A and 6B), we then examined which frequencies were most strongly tracked in the drum reconstructions. We focused on the 2–16 Hz, 4–32 Hz, and 8–64 Hz models because they produced the best reconstruction accuracies for drums and for the full rock envelope on average (Fig 6A). First, for each rock stimulus and model frequency range, the drum reconstructions were averaged across subjects. Then Welch's power spectral density (*pwelch* in Matlab) was computed using a Hamming window of 10 s with half-overlap. Then a null distribution of power spectral densities was created by shuffling the phases in each of the reconstructions, averaging the randomized reconstructions, and computing the power spectral density. This technique is based on methods to quantify magnitudes of peaks in frequency-following responses [48,49], where randomizing the phases of each signal and then averaging captures the spectra associated with the noise floor. 100 null spectra were computed for each

reconstruction. The average of these null spectra was subtracted from the true spectra to get the adjusted power spectral density for each model.

Next, we focused on the peak values in the adjusted power spectral density with respect to the expected rate of musical beats, since several studies have demonstrated that EEG and MEG tracks frequencies at multiples or fractions of this rate [35,47,67]. The musical beat rate (which we call the "tempo" here) was quantified using a beat-tracking algorithm [85]. This algorithm dynamically computes the timing of beats in a musical recording. The beat timings produced by the model were validated by the lead author of this study. The tempo was computed as the inverse of the median inter-beat interval.

Then, for each adjusted spectrum, peaks were identified with respect to 1 – 4x the music's tempo. For each scaling of the tempo, we identified the maximum value in the spectrum across all three models, using a frequency range ±8% around the multiple of the tempo [86].

## Supporting information

**S1 Fig.** (A) Envelope power spectra for each stimulus type prior to normalizing by the EEG power spectrum, as in Fig 1B. Lines indicate the median across stimuli of each type, and shaded regions indicate 95% quantiles of the distribution of 1000 bootstrapped median values. (B) Variance in the spectrum across trials. Because the speech trials all come from a single audiobook with one talker, they are more spectrally-similar to each other than the music stimuli.
(TIF)

**S2 Fig.** (A) Ridge regression was used to reconstruct the envelope from the EEG. Shown in the middle is the averaged reconstruction model across subjects and EEG channels for speech, where ridge regression was used to fit the model. Normally, the reconstruction model takes EEG as an input, but to simulate the spectral effects of the reconstruction model on the input, we have used broadband noise as the input in this example, which has a flat frequency spectrum. (B) When looking at the spectrum of the reconstruction with respect to a broadband noise input (black), it is clear that the reconstruction model accentuates certain frequencies and reduces higher frequencies (blue). However, it has no effect on the magnitude for frequencies corresponding to less than 2x the model width (1 Hz in this example), although it does add a delay that produces a phase shift at these frequencies (not shown). (C) The presence of low-frequency tracking has an effect on reconstruction accuracies. When the envelope and the EEG are highpass filtered by removing the moving average of the model width (500 ms), speech envelope reconstruction significantly drops, showing neural tracking at low frequencies untouched by the model. In contrast, reconstruction accuracies for the music stimuli significantly improves without these lower frequencies. Each color represents the testing reconstruction accuracies for one of the subjects.
(TIF)

**S3 Fig. Reconstruction accuracy curves for individual subjects.** (A) Z-scored reconstruction accuracies for each stimulus type (compare to Fig 3D). (B) Difference between z-scored reconstruction accuracy for speech and each of the other stimulus types (compare to Fig 3E). (C) Difference between stimulus-specific and stimulus-general reconstruction accuracy (compare to Fig 5A).
(TIF)

**S4 Fig.** (A) Envelope reconstruction accuracy based on Pearson's r, without the z-scoring used in the manuscript. (B) The difference in Pearson's r between the speech reconstructions and the reconstructions for each stimulus type shown. These were plotted identically to Fig 3D

and 3E, showing the median and 95% quantiles across trials and subjects.
(TIF)

**S5 Fig. We repeated the frequency-constrained reconstruction accuracy analysis on the Natural Speech dataset [35], to validate that the high low-frequency reconstruction accuracies we observed for speech were not specific to the current dataset (compare to Fig 3D).** Note that the Natural Speech dataset contained 20 trials of the audiobook, whereas the current dataset in the study only contained the first 6–7 trials (7 for most subjects, see S1 Table). (A) Shown are the reconstruction accuracies for all 19 subjects in the Natural Speech dataset, averaged across 20 trials. (B) We looked at reconstruction accuracies using all 20 trials of Natural Speech (blue, same results as A) and only the first seven trials (darker blue, dashed line in B). Wilcoxon's rank-sum test with Bonferroni correction for 16 comparisons was used to compare reconstruction accuracies between datasets; blue shows the comparisons with all 20 trials of Natural Speech, and darker blue shows comparisons is using just the first seven trials ($^{**}$ $p < 0.01$; $^{***}$ $p < 0.001$). In both instances, reconstruction accuracies were comparable to the current dataset and higher than the reconstruction accuracies for the other stimuli (see Figs 3D and S1). Note, however, that using all 20 trials produces above-chance reconstruction accuracies for the lowest frequency model, 0.0625–0.5 Hz. The reconstruction accuracies drop to chance when only seven trials are used. This indicates that the chance performance we observed in the current dataset may not be due to a low-frequency limitation on neural tracking of the speech envelope and may instead be a result of the limited amount of data in this study.
(TIF)

**S6 Fig. We optimized the hyperparameters of the PCA & spline model (specifically, the sampling frequency of the spline knots and the number of principal components) to a separate speech dataset, and we found that speech envelope reconstruction was better than music for all of the frequency ranges we examined (see Fig 3).** Here, we tested if music envelope reconstruction performs as well as speech if we optimize the hyperparameters for the music stimuli. (A) Using the same 500 ms model window as before, we found the optimal hyperparameter pairs for each stimulus type that maximized the average envelope reconstruction accuracy across subjects. These optimal hyperparameters were different than those found for Natural Speech (Rock = 16 Hz spline knots, 32 principal components (PCs); Orchestral = 16 Hz, 16 PCs; Vocals = 16 Hz, 32 PCs; Natural Speech = 32 Hz, 64 PCs). (B) We then computed the z-scored reconstruction accuracies (as in Fig 3) using these optimal hyperparameters. For each music stimulus, the dark blue dots on the left are the trial-by-trial reconstruction accuracies for all subjects using the Natural Speech hyperparameters (the same datapoints as those used to create Fig 3D), and the green dots on the right are using the music-optimized hyperparameters. The blue dots for the speech z-scored accuracies are based on the Natural Speech hyperparameters. Lines indicate the median values across trials and subjects. Even after optimizing the hyperparameters to the music stimuli, speech envelope reconstruction still outperforms music (Wilcoxon rank-sum relative to speech: $z_{rock} = 9.17$, $z_{orchestral} = 9.84$, $z_{vocals} = 7.16$, $p < 0.001$ for all comparisons).
(TIF)

**S7 Fig. We observed higher reconstruction accuracies for speech than the music stimuli (Fig 3D and 3E) but this could have been due to the higher cross-trial variability for the music stimuli than for the speech stimuli.** To control for this, we looked instead at within-trial reconstruction accuracy. (A) To get reconstruction accuracies for each trial, we split the trial into 10 evenly-sized folds, where each fold contained a random sampling of the data in

the trial. This was done in order to maximize the consistency in the EEG covariance and envelope spectrum across folds. Then models were fit on all trials with one fold left out and tested on the left-out fold. This was repeated 5 times using a new random sampling of folds each time, giving a total of 50 reconstruction accuracies (Pearson's r) for each trial. To get a null distribution of accuracies, the stimulus envelope was randomly circularly shifted, $1/10^{th}$ of the data was randomly sampled for testing, and the rest of the data was used for training. This was repeated 50 times to get 50 null reconstruction accuracies. (B) Because testing data is highly correlated with training data using this method, both the true and null reconstruction accuracies increase as lower frequencies are used for modeling. To correct this, we computed a d-prime reconstruction accuracy based on the distribution of true and null reconstruction accuracies. (C, D) Firstly, d-prime reconstruction accuracies dropped to zero for the 0.25–2 Hz model. This is a consequence of the reduced amount of data available in each trial; using lower-frequency models (with larger model windows) generated warnings in Matlab indicative of overfitting. But that aside, across all frequency ranges, d-prime reconstruction accuracy was significantly larger than all other music stimuli. Thick lines in D show significance of a permutation test comparing speech d-prime to each of the different stimulus types, p < 0.001 with Bonferroni correction for 24 comparisons. Plots (E and F) show the same results as C and D, respectively, for individual subjects. Overall, this indicates that, even when doing within-trial reconstructions to avoid the effects of cross-trial variance, speech is reconstructed better than the music stimuli.
(TIF)

**S8 Fig. To verify if the effects of reconstruction accuracy were a consequence of envelope reconstruction primarily based on eyeblinks (for example, if subjects inadvertently blinked at key times in the amplitude modulation), we repeated the envelope reconstruction using a single EEG component containing the subject's eyeblinks.** First, the EEG was highpass filtered by removing a moving average window 16 s long. Then the eyeblink component was calculated for each subject using independent components analysis (ICA; specifically, fastICA, as in [30]) and then identified empirically by the topography of the projection weights (transforming from the independent component to EEG space) and the time course of the EEG signal. One subject was left out because we could not reliably get a single component of eyeblinks using ICA. (A) Shown are the projection weights for this component averaged across the other 15 subjects. (B) Reconstruction accuracies using the eyeblink component. Thick lines indicate values significantly larger than zero based on a Wilcoxon signed-rank test with Bonferroni correction for 36 comparisons (p < 0.001). After repeating the envelope reconstruction analysis, we found that the reconstruction accuracies for all stimuli are still above chance at higher modulation frequencies, but considerably smaller than before (compare to Fig 3D). Similarly, the speech reconstructions were still significantly better than music. While this could indicate the involvement of eyeblinks, it is also plausible that the eyeblink component contains residual neural activity that tracks the envelope, since the topography for the eyeblink component overlaps the spatial weightings of the envelope reconstruction models (Fig 4). Thus, we constrained the analysis further by creating another input signal from the eyeblink component that only contained onsets at the peaks of the eyeblinks. (C) For each subject, the eyeblink component was highpass filtered again at 1 Hz by removing the moving average of a 1 s window, and then an eyeblink trigger was set individually for each subject to automatically identify eyeblinks by threshold crossing. The peak times of the eyeblinks were identified and the peak onset vector was used as input for envelope reconstruction. This ensured the envelope reconstruction would only use eyeblink timing and no other EEG activity contained in the eyeblink component. (D) When only the peak times were used, none of the reconstruction accuracies reached

a significance of p < 0.001 with Bonferroni correction (compare to B).
(TIF)

**S9 Fig. Temporal weights for reconstruction models for each individual subject, averaged across trials, after transforming to a "forward" model [83].** The models were converted from basis splines to delays, and then from principal components to EEG channels. The weights shown here were averaged across all 128 EEG channels (compare to Figs 4 and 5B).
(TIF)

**S10 Fig. Two different envelope reconstruction models were compared.** The "same-stimulus" model was trained and tested on the same stimulus type (this is identical to the "stimulus-specific" models from Fig 5 in the manuscript). The "cross-stimulus" models were trained on all of the presented stimuli for one stimulus type (the "Train" stimulus) and tested on each trial of another stimulus type (the "Test" stimulus). Shown here is the difference between the same-stimulus model and the cross-stimulus model. For example, the bottom left corner is the difference in reconstruction accuracy between a speech model trained on speech (same-stimulus) and a model trained on rock (cross-stimulus). Values greater than zero imply that the same-stimulus model outperforms the cross-stimulus model. Three asterisks (both white and black) indicate a significance of p < 0.001 for a Wilcoxon signed-rank test relative to a median value of zero after Bonferroni correction for 24 comparisons (this excludes the diagonals of each plot).
(TIF)

**S11 Fig. Based on the frontal topography of the weights that we observed for the 0.5–4 Hz model (Figs 4D and 5C), we were concerned that the greater reconstruction accuracies for the speech-specific model compared to the stimulus-general model (Fig 5A) might be related to eyeblink artifacts, which were especially prominent in some subjects (for example, if subjects unconsciously timed eyeblinks to envelope onsets in the stimulus).** While, to our knowledge, no eyeblink-based speech envelope reconstruction has been reported in the past, a 300–400 ms frontal negativity is indicative of eyeblink contamination in evoked response analyses [42]. (A) We examined the topography of the weights between 200–500 ms for each individual subject and found four subjects with topographies strongly indicative of eyeblinks. (B) After removing these subjects from analysis, however, the stimulus-specific model for speech still outperformed the stimulus general model for 0.5–4 Hz (Wilcoxon signed-rank test with Bonferroni correction for 32 comparisons, p < 0.001), but this was no longer true for the 1–8 Hz model. (C and D) Additionally, the time course and topographies of the model weights were very similar to what was observed in our analysis using all 16 subjects (compare to Figs 4 and 5).
(TIF)

**S12 Fig. $R^2$ values between the stimulus-specific and stimulus-general models, without any scaling or shifting of the stimulus-general model.** As in Fig 5E, this is shown for the 4–32 Hz and 0.5–4 Hz models. The y-axis has been restricted to a range from -1 to 1 for easier comparison of the medians (lines); datapoints below $R^2$ are not shown.
(TIF)

**S13 Fig. Rock instrument reconstruction accuracies for individual subjects.** (A) and (B) are plotted identically to Fig 6A and 6B respectively.
(TIF)

**S1 Table. List of stimuli used in this experiment, including stimulus duration and the number of times it was presented across all 16 subjects.**
(XLSX)

## Acknowledgments

We would like to thank Pedro Marquez for some earlier analyses of this dataset, which inspired the main questions of this study. We would also like to thank Giovanni Di Liberto, Aaron Nidiffer, Andrew Anderson, Michael Broderick, Shyanthony Synigal, and the others in the Lalor Lab for their comments and corrections on this manuscript.

## Author Contributions

**Conceptualization:** Nathaniel J. Zuk, Jeremy W. Murphy, Edmund C. Lalor.

**Data curation:** Jeremy W. Murphy, Edmund C. Lalor.

**Formal analysis:** Nathaniel J. Zuk.

**Funding acquisition:** Richard B. Reilly, Edmund C. Lalor.

**Investigation:** Nathaniel J. Zuk, Jeremy W. Murphy.

**Methodology:** Nathaniel J. Zuk, Jeremy W. Murphy.

**Project administration:** Richard B. Reilly, Edmund C. Lalor.

**Resources:** Richard B. Reilly, Edmund C. Lalor.

**Software:** Nathaniel J. Zuk.

**Supervision:** Richard B. Reilly, Edmund C. Lalor.

**Validation:** Nathaniel J. Zuk, Edmund C. Lalor.

**Visualization:** Nathaniel J. Zuk.

**Writing – original draft:** Nathaniel J. Zuk.

**Writing – review & editing:** Nathaniel J. Zuk, Jeremy W. Murphy, Edmund C. Lalor.

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
