## [Decision Letter · Decision Letter 0]

21 Mar 2021

Dear Dr Zuk,

Thank you very much for submitting your manuscript "Envelope reconstruction of speech and music highlights unique tracking of speech at low frequencies" for consideration at PLOS Computational Biology.

As with all papers reviewed by the journal, your manuscript was reviewed by members of the editorial board and by several independent reviewers. In light of the reviews (below this email), we would like to invite the resubmission of a significantly-revised version that takes into account the reviewers' comments.

Dear Authors,

As you will see reviewers 2 and 3 have raised serious issues re both the novelty and the significance of the results even though all three reviewers appreciated your methodological contributions. Reviewer 2 is asked whether the effect that you observe is not a somewhat trivial consequence of the differences in. temnporal mod spectra of the speech vs music stimuli. I believe that their concerns are reasonable. I hope that you will be able to address them.

Best wishes,

F. Theunissen

We cannot make any decision about publication until we have seen the revised manuscript and your response to the reviewers' comments. Your revised manuscript is also likely to be sent to reviewers for further evaluation.

Sincerely,

Frédéric E. Theunissen

Associate Editor

PLOS Computational Biology

Wolfgang Einhäuser

Deputy Editor

PLOS Computational Biology

Dear Authors,

As you will see reviewers 2 and 3 have raised serious issues re both the novelty and the significance of the results even though all three reviewers appreciated your methodological contributions. Reviewer 2 is asked whether the effect that you observe is not a somewhat trivial consequence of the differences in. temnporal mod spectra of the speech vs music stimuli. I believe that their concerns are reasonable. I hope that you will be able to address them.

Best wishes,

F. Theunissen

Reviewer's Responses to Questions

**Comments to the Authors:**

Reviewer #1: The authors explored whether the human brain tracks speech and music differently using EEG. Using a novel method that enabled them to examine envelope tracking in narrow frequency bands, they reported that speech was better tracked at all the frequency range they examined. Furthermore they found a different topography in weightings for speech-specific model reconstruction compared to other stimulus-general or stimulus-specific model reconstruction at the low frequency range (0.5-4 Hz). Overall, the MS is very well written and clear; the methods are sound. I do not have specific major concerns but I would like the authors to be clearer on the following points:

1. The very low frequency range is particularly interesting but there are a few gaps in the MS in my mind. How good is the BioSemi EEG system sampling at very low frequency (~0.1 Hz)? What is hard to ascertain is that whether the noise floor is too big (either because of the equipment constraints or not having enough trials, c.f. Fig. S3) to conclude that tracking speech is better at "all" frequencies. The analysis of the drum track is an interesting analysis to examine low frequency envelope tracking. But it is also debatable one would track every beat of the drum (during passive or even active listening). Rather, one suspects that you would care mostly the "strong" beat of each measure and acoustically, the relative magnitude would also be different on this beat. This is significant because instead of looking at the harmonics of these beats, one would argue the sub-harmonics would be more important. Thus for three songs mentioned in ll 282-283 with beats being below 2 Hz (i.e., less than 120 beats per minute; bpm), one could argue that listeners would be following at 1/4 of the rate (if there are 4 beats to a measure, i.e., 0.5 Hz). If the music is in a slower pace, it can greatly reduce to 0.1-0.2 Hz range. Thus, it would be useful to keep the frequency axis to show up to ~0.1 Hz range, e.g., Fig. 6C-E and Fig. S5 so that one could claim that speech tracking is much better than music for "all frequencies." If the very low frequencies are not conclusive, it's better to leave that as a caveat to be investigated in future studies.

2. The parietal weighting difference in the speech-specific reconstruction is interesting, but the reader is left to wonder the greater significance of this besides that it is different from the stimulus-general reconstruction models and it is different from other stimuli. Are there any reasons why this difference would be parietal of origin?

3. Since this paper would be of interest to researchers that are interested in music processing, it would be nice to have the musical terms to be more precise. The term "classical" is loaded. I think how it is used in this MS is the colloquial reference of instrumental music (that's of the Western Tonal Tradition) as well as it not being contemporary. Table S1 suggests that there are actually film music included and it's contemporary. I think using the term "orchestral / instrumental music" would be more appropriate. (It depends on whether the piano version or the orchestral version of Claire de Lune was used and if it's not piano, you can drop "instrumental" all together).

Minor comments:

Ln 100: rephrase "we expected that music may track speech comparably". You probably meant "music enveloping tracking would be comparably to that of speech" or to that effect.

Fig. 5F: circular shift plots: What are the units for the vertical axis (cycle?) and can one contextualize what those values mean in a circular shift? Perhaps more details to guide the readers would be beneficial.

Reviewer #2: This study compares envelope coding in EEG activity during passive listening to speech and music. The authors use a novel stimulus reconstruction approach that band-limits each model to operate within a 3-octave range in the neural data, with separate models trained and tested across different bands that range from phrase-level modulations (<0.5Hz) to putative pitch-tracking (>32Hz). Based on prior work, the authors hypothesize that envelope tracking across music and speech may be comparable at higher modulation frequencies above 2Hz. The notable differences in modulation spectra that exist between the different stimulus categories are acknowledged, and the null hypothesis is that relative reconstruction accuracies across categories directly reflect these stimulus differences. Using within-category models (trained and tested on each stimulus independently) the results are broadly consistent with the null hypothesis, whereby speech reconstruction outperforms music across all frequencies. Slight differences are found in the modulation frequency at which peak accuracy occurs (lower for speech than music). To disentangle general-purpose auditory coding from domain-specific mechanisms, models are then trained/tested using data pooled across stimuli and compared with the performance of stimulus-specific models. While general vs. specific models yield similar performance at higher frequencies, speech-specific models are found to outperform the general models at lower frequencies (0.5-4Hz), which is driven by a subset of parietal electrodes. The authors use this finding to claim that “below 1Hz mechanisms exist that specifically track speech”.

On the whole I enjoyed reading this paper. The manuscript is well written, the data seems appropriately collected, analyses are mostly effective in testing the hypothesis, and statistical tests are appropriate. However, given the nature of the differences in modulation spectra between stimuli, I have major concerns with the ultimate claim that low-frequency envelope tracking is specific to speech (see below). Because of this concern, the novelty of the current study is largely methodological rather than theoretical. It is my opinion that publication in PLoS Comp Bio is warranted if the authors significantly dial back their claims of speech-specificity and address the specific issues listed below.

Major:

My main concern relates to the degree to which envelopes of speech differ from those in each music category. Figure 1A shows examples of slow modulations in speech (presumably sentence/phrase structure that results in the 0.5Hz peak). This modulation structure is virtually absent in the two instrumental categories, and only minimally present in vocals. This all seems intuitive - a rock song isn’t likely to be interspersed with silences in the same way an audiobook is.

This is problematic in two ways:

1. Firstly, at these very low modulation rates, if the signal is coming primarily from speech and not music - is it really surprising that the speech-specific model outperforms the stimulus-general model (the latter would essentially just have added noise due to low SNR of music)? This could also be the case if the neural response to power in a given modulation band wasn't linear across the range of stimulus-space occupied by both music and speech.

2. Secondly (and perhaps more critically), the current study assumes that a fixed encoding mechanism exists across all frequencies. However, there may be two distinct mechanisms taking place for encoding envelope modulations at low and high frequencies respectively. This is what is suggested by ECoG research (e.g. see Hamilton et al., 2018) in which distinct neural populations in the STG respond to phrase-level onsets, independent from populations that encode the faster modulations in the continuous stream. Ultimately then, there may exist two different envelope mechanisms that are both domain-general. However, since the low-frequency structure only exists for speech within the current stimuli, encoding of slow envelope modulations are simply masquerading as speech-specialized.

Minor:

* Introduction line 54: In the two papers cited (Oganian & Chang, 2019; Kojima et al., 2020), evoked responses are time-aligned to envelope edges which mark vowel-nucleus onsets, not syllable onsets. Although the two are highly correlated they have different theoretical implications.

* Figure 1B: Because "1/f" effects dominate the modulation spectrum, it's difficult to visually extract what the critical differences are between the four stimuli (aside from speech having a uniformly higher power). I suggest including a panel that plots normalized spectra for each music condition relative to speech (in the same vein as the reconstruction accuracies in figure 3E).

* Lines 100-111: a general comment about the flow - at this point in the paper it is unclear how the narrowband modeling approach allows for a clear dissociation between the competing hypotheses. Only when stimulus general vs. specific models are compared can one infer anything about higher-order effects above and beyond what would be expected if neural differences just reflected acoustic ones. As the results currently read, this paragraph sounds like the narrowband modeling allows for such a dissociation.

* The spline basis transform needs to be further motivated. What are the specific benefits of using such an approach above other methods? I am unfamiliar with this technique (and it’s safe to assume that most readers will also be). The transform seems to serve dual purposes of smoothing the PCs while also imposing a high frequency cut-off via the selection of knot sampling frequency. If these are the primary objectives, why not use more conventional smoothing and bandpass filtering techniques? There is mention of the need to control for temporal correlations but no explicit description of how splines may achieve this (or why temporal collinearity needs to be controlled if time-lags are included as features in the models).

* The collective effect of moving-average subtraction and spline transformation is a bandpass on the EEG data (figure 2C). However, on the stimulus end, only removal of the moving-average is applied. Theoretically this means that lower-frequency neural modulations could map onto higher-frequency envelope power? A brief explanation of why this is/isn’t the case is needed.

* Figure 5: Y-axis labels are missing from 5E-F (the values seem rather high to be the actual R^2 from correlating predicted vs. actual envelope, but perhaps this is because predictions are evaluated on a trial-by-trial basis?). More generally, the results shown in figure 5E-F are not immediately easy to square with the results of 5A. Given that the speech-specific model outperforms the general at low frequencies, why aren’t the general-to-specific model fits lower in this condition? If the scaling factor and circular shift is able to compensate for this difference, the results need to be visualized such that this becomes clear (e.g. show the unscaled and unshifted R^2 side-by-side with current data).

* Relatedly, it's unclear why music-specific models (particularly rock) would exhibit differences from general models that actually hurt rather than improve performance across all frequencies. Authors should, at the least, include in the discussion some speculation as to why this may be (is it just a data limitation issue?).

Miscellaneous:

* line 189 refers to figure 1b (not figure 2b).

Reviewer #3: Zuk and co-workers present a thorough study on the envelope tracking of speech and music measured with EEG. The authors paid particular attention to potential biases on the reconstruction performance due to different stimulus types and thus added two additional processing steps to establish their envelope reconstruction models (PCA + spline basis). The results showed superior reconstruction of speech stimuli over music types (rock music, vocals extracted from rock pieces, classical music) that is presented to be unique to low frequencies and parietal channels.

I appreciate the extended analyses as well as the care that has been taken during control analyses and in interpreting the results. I also think that the study adds valid and important methodological points of discussion to the field of envelope tracking of speech and music. However, I think that the main empirical finding(s) of this study could be better communicated. As is, the manuscript reads very analysis-heavy and undermines the impact of the empirical results. In addition, these main empirical findings require further tests.

Please find these and further suggestions in the main and minor points below.

Main points:

1. The main conclusion of the paper mentioned in the title and abstract is that speech is tracked ‘uniquely’ by low frequencies. These results mainly stem from section “Speech envelope tracking at low frequencies […]” (l.202-268) and are based on results presented in Figure 5, which aims to answer the question on whether model “differences affect neural tracking of the envelopes for the different stimuli” (l.219/220) and whether “speech tracking at low frequencies [is] a result of common processing across stimulus types” (l.223/224). This analysis compares the performance of the ‘specific’ model (training data are of the specific stimulus class) to the ‘general’ model (training data are trials across stimulus types) that is assumed to “capture consistent trends across all stimulus types relevant to envelope tracking” (l.227). This would hold only if each trial has a similar impact on models. However, as explained in this paper, music stimuli are more variable and thus might not affect models as much as compared to speech stimuli that show less variance. I suggest performing a cross-decoding/reconstruction analysis that would use a model of one stimulus type (e.g., speech) to reconstruct trials of the other stimulus types. Thus, the results would be similar to a confusion matrix where the specific models would lie on the diagonal. This way the questions above will be more directly answered and also the main conclusion of unique tracking of speech at low frequencies (Fig 5A) can be better examined.

2. The other main conclusion that speech tracking was “uniquely associated with increased weighting of parietal channels” (l.202/3, also in abstract) is based on descriptive results presented in Fig 4d, lower plots. I do agree that these results are an indication but if this is to be a main finding, I suggest to statistically test whether the weighting of these channels is higher compared to the other stimulus types. If the statistics over model parameters are not straight-forward here, the permutations (null models) might provide a good normalization (similar to analysis of reconstruction accuracy (Fig 3D,E). Related to this, I wonder how this finding is interpreted.

3. Please be clearer on the reason of using the unusual model using PCAs and a spline basis for envelope tracking. It is the main model used in this study and represents one major novelty. However, it is more complicated than previous envelope tracking approaches. For example, it would strengthen the paper to explain the advantages of 1) results of envelope tracking with different windows sizes and EEG filtering in comparison to the PCA-spline approach and 2) why it overcomes stimulus-specific acoustic regularities of the different stimulus types clearly in the main text. The latter is not clear to me and I do not follow how this analysis manages to abstract from inherent acoustics of stimuli affecting envelope tracking. Related to this point: In figure 2B, the ‘standard’ EEG envelope tracking provides higher reconstruction accuracy in comparison to all but one model type of the new approach and might make readers wonder about this tracking approach.

Minor points:

1. It seems that the silence in sounds was not controlled. It’s important to check whether there are differences in this aspect between sound types. The examples in Fig 1A suggest for example that speech has longer pauses than other types. This could be an explanation of the superior tracking as for speech. As this cannot be changed, I suggest providing a statistic on silences/pauses (length and #occurences) for the different stimulus types. If differences are found, this should enter the discussion of the results.

2. l. 236-240: I very much appreciate that potential contributions of eye movement on reconstruction accuracies are openly discussed and that the comparison of specific vs general model were reanalysed. However, as these did change the outcome of this analysis, I’m also wondering about the remaining results (I can imagine that speech but especially music might lead to movement locked to the envelope). Performing the same analyses after preprocessing dedicated to remove EEG associated with EOG or other non-brain sources could resolve this (see for example Zuk et al., 2020). Given that backward modelling of long, continuous sounds, EOG and other artifacts might not have a strong impact if blinks or movements are unrelated to the stimulus. Thus another way of showing that results do not depend on artifacts would be to examine whether eye movements occurred at specific times during the sound.

3. l.199-200. I do not follow this sentence. In contrast to what is stated, it was mentioned earlier that speech envelopes had lower cross-trial spectral variance (l. 190). I might miss a the differences here but it would be good to provide to which result of this section this suggestion is related to.

4. l. 289: Figure 2b -> 1b?

5. l.221 “speech tracking is better at low frequencies (0.5 – 4Hz) than music tracking […]”. Maybe I missed an important part but I understood that one main finding of the first section (Fig 3) was that speech envelopes are tracked better across all but the very lowest frequency range? I thus wonder why the emphasis was put on the 0.5-4Hz range (it was reported as the peak difference between speech and music stimuli, l.175 but all other bands were very similar).

6. Fig5E. to be consistent with 5F, indicating significances should be included

7. l. 326-329. The fact that there are multiple sound sources (instruments) could lead to attention switches (between instruments and between single instruments and the aggregate music) and as such affect tracking that is shown to depend on attentional focus and state. The fact that the drums are tracked best and similarly well to the overall rock stimulus does not imply that paying attention to an isolated instrument would lead cannot approach speech tracking. As such, this might (partially) explain the superior speech tracking, which consisted of one source in isolation vs. a multi-source sounds for rock music.

8. Fig. 2B. Could you please provide an explanation (response letter suffices) on why the full model (128 PCs, no spline) performed worse than EEG model? Using all principal components should provide the same amount of information as the original EEG data. It might thus be due to the specific model type.

9. Fig. 4C, D and Fig 5C: Please add information on how the intervals of the presented weight topographies were chosen (for example, according to weight time-courses (e.g., delay minimum weight +/-20ms)

10. l.43: don’t -> do not

11. l.588: figure 2E does not exist, my guess is that this refers to Fig 1C.

**Have all data underlying the figures and results presented in the manuscript been provided?**

Reviewer #1: Yes

Reviewer #2: Yes

Reviewer #3: Yes

PLOS authors have the option to publish the peer review history of their article (what does this mean?). If published, this will include your full peer review and any attached files.

Reviewer #1: **Yes: **Adrian KC Lee

Reviewer #2: No

Reviewer #3: No
---

## [Editor Report · Decision Letter 1]

21 Jul 2021

Dear Dr Zuk,

Thank you very much for submitting your manuscript "Envelope reconstruction of speech and music highlights stronger tracking of speech at low frequencies" for consideration at PLOS Computational Biology. As with all papers reviewed by the journal, your manuscript was reviewed by members of the editorial board and by several independent reviewers. The reviewers appreciated the attention to an important topic. Based on the reviews, we are likely to accept this manuscript for publication, providing that you modify the manuscript according to the review recommendations.

Dear Authors,

I have read your paper and your response to the reviewers. I have a few comments of my own. I appreciate the care you took to address the reviewers comments. Also you are trying to demonstrate something that is not easy. It is very hard for one to accept that speech is special when the other stimuli have such different modulation power (and clearly smaller as well shown in your figure 1B!) Much of your effect could just be do to SNR.. You work hard to show that it is more than that by examining the value of the weights and comparing goodness of fits to expected null values. One could argue that the best stimulus for the comparison would have match amplitude envelope spectrum but maybe lacking linguistic. But then one could also argue that such stimuli are not natural and, thus, not engaging the subjects.. In any case, I appreciate the effort you made by remain somewhat unconvinced about the effect size. The drumming control was clever and the small boost unexpected and clearly in favor of your argument. But maybe a rock musician involved in playing with the band could have EEG signals that are just as entrained...

I believe that your results speak for themselves and in the latest version you are indeed more careful in your statements. Here are some minor points that I would like you to address, maybe just by sending me response.

1. In terms of the novelty of your method: I am impressed by your overall rigor and I also appreciate that you are able to perform an analysis in octave bands. But its seems like you could have gotten the same results just by estimating the coherence and coherency (to look at systematic errors in phase) instead of r or R2 for your measure of goodness of fit - and it would give you the goodness of fit as a function of frequency after just fitting a regular regularized linear model. See for example (Hsu A, Borst A, Theunissen FE. Quantifying variability in neural responses and its application for the validation of model predictions. Network. 2004;15(2):91-109). Would the results be different?

2. I would use the same yscale in fig 4A. Maybe -20 to 5 ? Also the very low band would show greater weights for Vocal than speech- no? It might be an interesting comparison

3.. The fact that you get better reconstruction with the general model suggests that you are data limited and just more so for the stimuli other than speech almost certainly because it has a higher SNR. I think you address some of this in 5E by estimating the chance R2 distribution but I also think that it is hard to be completely convinced. I don't really know how to correct for differences in SNR when one is also using regularization. I know that you use the same hyper-parameters in your data analysis and so maybe it could just involve restricting the data set of the speech to match some form of integrated SNR? If you have addressed this, I did not completely follow your argument in the paper.

Sincerely,

Frédéric E. Theunissen

Associate Editor

PLOS Computational Biology

Wolfgang Einhäuser

Deputy Editor

PLOS Computational Biology

[LINK]

Dear Authors,

I have read your paper and your response to the reviewers. I have a few comments of my own. I appreciate the care you took to address the reviewers comments. Also you are trying to demonstrate something that is not easy. It is very hard for one to accept that speech is special when the other stimuli have such different modulation power (and clearly smaller as well shown in your figure 1B!) Much of your effect could just be do to SNR.. You work hard to show that it is more than that by examining the value of the weights and comparing goodness of fits to expected null values. One could argue that the best stimulus for the comparison would have match amplitude envelope spectrum but maybe lacking linguistic. But then one could also argue that such stimuli are not natural and, thus, not engaging the subjects.. In any case, I appreciate the effort you made by remain somewhat unconvinced about the effect size. The drumming control was clever and the small boost unexpected and clearly in favor of your argument. But maybe a rock musician involved in playing with the band could have EEG signals that are just as entrained...

I believe that your results speak for themselves and in the latest version you are indeed more careful in your statements. Here are some minor points that I would like you to address, maybe just by sending me response.

1. In terms of the novelty of your method: I am impressed by your overall rigor and I also appreciate that you are able to perform an analysis in octave bands. But its seems like you could have gotten the same results just by estimating the coherence and coherency (to look at systematic errors in phase) instead of r or R2 for your measure of goodness of fit - and it would give you the goodness of fit as a function of frequency after just fitting a regular regularized linear model. See for example (Hsu A, Borst A, Theunissen FE. Quantifying variability in neural responses and its application for the validation of model predictions. Network. 2004;15(2):91-109). Would the results be different?

2. I would use the same yscale in fig 4A. Maybe -20 to 5 ? Also the very low band would show greater weights for Vocal than speech- no? It might be an interesting comparison

3.. The fact that you get better reconstruction with the general model suggests that you are data limited and just more so for the stimuli other than speech almost certainly because it has a higher SNR. I think you address some of this in 5E by estimating the chance R2 distribution but I also think that it is hard to be completely convinced. I don't really know how to correct for differences in SNR when one is also using regularization. I know that you use the same hyper-parameters in your data analysis and so maybe it could just involve restricting the data set of the speech to match some form of integrated SNR? If you have addressed this, I did not completely follow your argument in the paper.

Figure Files:

Data Requirements:

Reproducibility:

References:

---

## [Editor Report · Decision Letter 2]

18 Aug 2021

Dear Dr Zuk,

We are pleased to inform you that your manuscript 'Envelope reconstruction of speech and music highlights stronger tracking of speech at low frequencies' has been provisionally accepted for publication in PLOS Computational Biology.

Best regards,

Frédéric E. Theunissen

Associate Editor

PLOS Computational Biology

Wolfgang Einhäuser

Deputy Editor

PLOS Computational Biology

Dear Authors,

Thanks for responding to my comments. I still think that a coherence analysis might be a more intuitive way to get at the same "frequency-dependent" effect but other researchers can try to compare your approach to a coherence based approach.

Best wishes,

Frederic.

---

## [Editor Report · Acceptance letter]

14 Sep 2021

PCOMPBIOL-D-21-00206R2 

Envelope reconstruction of speech and music highlights stronger tracking of speech at low frequencies

Dear Dr Zuk,

I am pleased to inform you that your manuscript has been formally accepted for publication in PLOS Computational Biology. Your manuscript is now with our production department and you will be notified of the publication date in due course.

With kind regards,

Andrea Szabo
